# THE COGNITIVE BANDWIDTH BOTTLENECK: SHIFTING LONG-HORIZON AGENT FROM PLANNING WITH ACTIONS TO PLANNING WITH SCHEMAS

## ABSTRACT

Enabling LLMs to effectively operate long-horizon task which requires long-term planning and multiple interactions is essential for open-world autonomy. Conventional methods adopt planning with actions where a executable action list would be provided as reference. However, this action representation choice would be impractical when the environment action space is combinatorial exploded (e.g., open-ended real world). This naturally leads to a question: *As environmental action space scales, what is the optimal action representation for long-horizon agents?* In this paper, we systematically study the effectiveness of two different action representations. The first one is conventional planning with actions (PwA) which is predominantly adopted for its effectiveness on existing benchmarks. The other one is planning with schemas (PwS) which instantiate an action schema into action lists (e.g., "move [OBJ] to [OBJ]" → "move apple to desk") to ensure concise action space and reliable scalability. This alternative is motivated by its alignment with human cognition and its compliance with environment-imposed action format restriction. We propose cognitive bandwidth perspective as a conceptual framework to qualitatively understand the differences between these two action representations and empirically observe a representation-choice inflection point between ALFWorld (∼35 actions) and SciWorld (∼500 actions), which serve as evidence of the need for scalable representations. We further conduct controlled experiments to study how the location of this inflection point interacts with different model capacities: stronger planning proficiency shifts the inflection rightward, whereas better schema instantiation shifts it leftward. Finally, noting the suboptimal performance of PwS agents, we provide an actionable guide for building more capable PwS agents for better scalable autonomy.

## 1 INTRODUCTION

Humans often make decisions by instantiating abstract action templates into concrete, executable steps, a process we term planning with schemas (PwS) (Schmidt, 1975). In open-ended, real-world settings where the action space grows combinatorially, we do not search directly over executable primitives; instead, we ground a template such as "move [OBJ] to [OBJ]" into a specific action like "move apple to desk". This schema-driven mechanism supports generalization from simplified settings to rich environments with vast or even unbounded action spaces. With recent advances in the planning capabilities of Large Language Models (LLMs) (OpenAI, 2023; DeepSeek-AI et al., 2025; Bai et al., 2025), numerous LLM-based autonomous agents have been proposed to tackle long-horizon problems. For example, an agent tasked with characterizing an unknown substance must decompose a high-level goal into a coherent sequence: acquire the sample, prepare the apparatus, run the experiments, and synthesize a conclusion. Methodologies such as fine-tuning (Xi et al., 2025; Zhang et al., 2025; Shinn et al., 2023) and advanced prompting (Zhao et al., 2024a; Wang et al., 2024) enable these agents to execute plans over multiple interactions more effectively.

However, different from human cognition, most existing agents are designed for **planning with actions (PwA)**, where the environment provides a list of executable actions from which the agent selects one at each step. While effective in many benchmarks, this paradigm faces a scaling challenge: in complicated environments, the action list becomes intractably long or even innumerable

(e.g., open-ended real scenarios). This overload not only strains the context window but, more critically, creates a decision-making bottleneck. This leads to a foundational question: *As environmental action space scales, what is the optimal action representation for long-horizon agents?*

To answer this question, we compare two competing paradigms. The first is the conventional PwA approach, which dominates current research and performs well in existing evaluation setups. The second is our proposed alternative, **planning with schemas (PwS)**, which has been comparatively underexplored. Another motivation for this alternative choice, aside from the cognitive perspective, is that interactive environments typically require actions in a structured format. Free-form generation often violates this constraints, whereas schema-based planning treats action selection as instantiating vetted templates, producing interface-conformant outputs that scale more reliably. We systematically compare these paradigms across varying action-space sizes and seek to understand the drivers of the *representation-choice inflection point* where the optimal action representation switches.

To formalize the trade-offs between paradigms, we introduce the **Cognitive Bandwidth Perspective**: **a given LLM possesses a fixed cognitive bandwidth, which PwA and PwS allocate differently**. This is a conceptual scaffold for reasoning about load allocation rather than a quantitative probe of model capacity. Under PwA, the model bears a heavy **Environment Understanding (EU)** burden to interpret noisy observations and parse long action lists while under PwS, this burden shifts to **Schema Instantiation (SI)**, which demands reasoning to ground templates into valid actions.

Our experiments across four environments of increasing action space scale, TextCraft, WebShop, ALFWorld, and SciWorld,validate the existence of performance inflection. We observe a critical *representation-choice inflection point*: in low-to-medium action spaces (ALFWorld$\sim$35 actions), PwA outperforms PwS by 33.4% on average, as SI overhead dominates. However, in environments with lengthy action spaces (SciWorld$\sim$500 actions), this trend inverts. PwS achieves an 8.1% average advantage as EU load becomes prohibitive. **Thus, the optimal action representation is not universal. PwA suffers from action space scaling and PwS being convincing for scalability.**

To further investigate the mechanisms driving this inflection, we design *cognitive-load stress tests* that incrementally inject distractor actions into the action list. This perturbation enlarges and adds noise to the action space, increasing EU burden without altering task semantics, helping us to study the dynamic of inflection point regarding different models. We find that the location of the inflection point for specific model is affected by two factors: (i) the model's *agentic proficiency*—its ability to perform agentic planning, proxied by PwA success on ALFWorld without distractors; and (ii) its *schema-instantiation capability*, the efficiency with which templates are grounded, evidenced by PwS success. Our results support that **Models with strong agentic proficiency degrade more slowly as the list grows, shifting the inflection rightward whereas models with strong SI proficiency (e.g., >25% under schemas) incur lower SI overhead, shifting the inflection leftward.**

Finally, by recognizing how the inflection point varies with model capability and that PwA is constrained by intractably long action lists, we offer empirically grounded guidance for strengthening PwS and moving the inflection left. Examining contemporary reasoning formats and post-training recipes, we observe that **(i) extending reasoning depth, as in large reasoning models, is generally helpful but not decisive for PwS when SI remains the bottleneck**, and **(ii) post-training that emphasizes multi-turn tool use reduces SI load and shifts the inflection point left.**

Our contributions are threefold:

- We systematically study the optimal action representation problem under environmental action-space scaling and propose Cognitive Bandwidth Perspective to provide a conceptual framework for understanding trade-off between different action representations.

- A representation-choice inflection point is discovered empirically, providing evidence for the scaling limitations of conventional planning with actions (PwA) and motivates the adoption of planning with schemas (PwS) for more scalable action representation.

- We design cognitive load stress test to understand the interplay between model capacity and inflection point and thereby provide insightful recommendations for developing more capable schema-based agents which could be more reliable to action-space scaling.

These findings establish an alternative perspective for scaling autonomous agents to increasingly complex real-world environments. We will make our code open-sourced to foster future research.

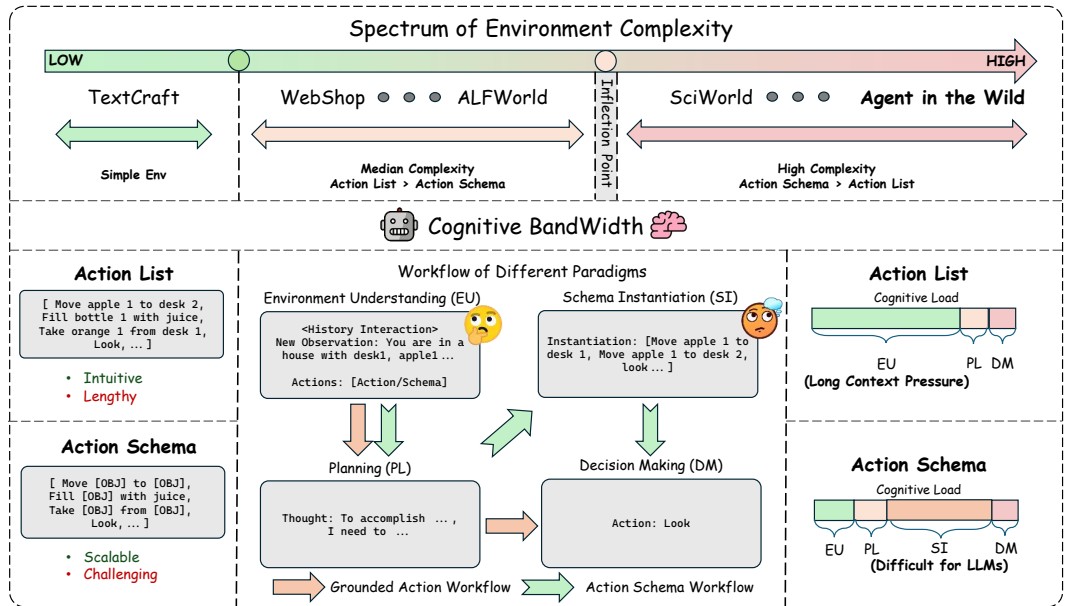

Figure 1: The relationship between environment complexity and agent paradigm effectiveness. In simple environments, Grounded Actions suffice. As complexity grows past an "inflection point", the Action Schema paradigm becomes superior by shifting the cognitive load from processing long action lists (high EU load) to a challenging but scalable Schema Instantiation (SI) step.

## 2 RELATED WORK

**LLM-based agents** Prior work on constructing LLM-based agents can be broadly classified into following categories. **1. Prompting and In-Context Learning.** These methods leverage instructions to steer LLM behavior, leading to improved task performance via explicit long reasoning (Yao et al., 2023; Wei et al., 2022), self-reflection to learn from previous errors (Renze & Guven, 2024; Shinn et al., 2023), long-term planning to organize action sequence (Nayak et al., 2024; Prasad et al., 2024; Sun et al., 2023), role-playing to incentivizing LLM's reasoning with personas (Xu et al., 2024), and multi-agent debate to spur agent's collective intelligence (Xu et al., 2025a; Du et al., 2024). This group as methods are simple but effective, exhibiting great transferability across domains. **2. Training Methods.** These approaches adapt the model's parameters using supervised fine-tuning (Yuan et al., 2025; Qiao et al., 2024; Xi et al., 2024) where a explicit "golden trajectory" is provided for agent to immitate or reinforcement learning (Jin et al., 2025; Wang et al., 2025b; Fang et al., 2025) where the agent learn by getting reward from the environment, gradually optimize its own policy. While often achieving high performance on specific benchmarks, this paradigm is resource-intensive and the resulting models prune to PwA paradigm, facing scaling challenge.

**LLMs in Long Context.** While current LLMs exhibit extraordinary capability on short-context problems, they falter on tasks requiring synthesis of information across long historical context in either text modality (Zhang et al., 2024; Wu et al., 2025) or multimodality setup (Wang et al., 2025a; Maharana et al., 2024). One research direction involves multi-agent frameworks (Liu et al., 2025; Zhao et al., 2024b; Li et al., 2025a) multiple agents work collaboratively. Another approach is to memorize critical information via either reinforcement learning (Zhou et al., 2025; Yu et al., 2025) or constructing information database (Chhikara et al., 2025; Xu et al., 2025b; Li et al., 2025b), where a external memory is provided to assist agent reasoning.

## 3 METHODS AND COGNITIVE BANDWIDTH PERSPECTIVE

Understanding bottlenecks in multi-turn model–environment interaction is nontrivial. To provide a systematic perspective of analyzing long-horizon agents, we introduce the Cognitive Bandwidth Perspective, a conceptual framework that qualitatively decomposes agent workflows into distinct stages

to diagnose performance limits. This section details our methodology for deriving action schemas from action lists and formalizes Cognitive Bandwidth Perspective for following investigations.

## 3.1 DERIVATION OF ACTION REPRESENTATIONS

**Executable Action List.** For ALFWorld and SciWorld, the executable action list is obtained directly from the environment at each timestep, following standard practice. For WebShop, which does not natively provide such a list, we generate it by parsing the observation to extract all currently clickable elements. Finally, for the inherently schema-based TextCraft environment, we choose to only study the schema-based paradigm due to the relatively low difficulty of the environment, evidenced by satisfying performance over various LLMs.

**Abstract Action Schema Derivation.** For TextCraft and WebShop, the action schemas (e.g., craft [num] [ITEM], click [BUTTON]) are intrinsic to the environments' design and were used directly.

For ALFWorld and SciWorld, we derive the schemas by abstracting the complete grounded action lists provided by their environments. Specifically, we replace all object arguments in the actions with a generic [OBJ] token and collect the set of unique resulting templates. We provide the grounded action list size and the length of action schemas in Table 1. The comprehensive list of the action schemas used in our experiments is available in Appendix A.

Table 1: The correlation between action space size and environment complexity. Larger action space indicates more complex environment.

| Environment | TextCraft | WebShop | ALFWorld | SciWorld |
|---|---|---|---|---|
| Action List Length | - | 3∼10 | 30∼40 | 400∼600 |
| Schema List Length | ∼3 | 2 | 11 | 26 |

## 3.2 COGNITIVE BANDWIDTH PERSPECTIVE: A CONCEPTUAL ANALYTIC FRAMEWORK

To investigate the reasoning obstacles for long-horizon agents under different paradigms, we introduce **Cognitive Bandwidth Perspective** as a conceptual analytic framework for diagnosing how cognitive load is distributed across stages. This perspective posits that an LLM agent's performance is fundamentally constrained by its intrinsic processing capacity (e.g., cognitive bandwidth), which must be sufficient to handle the cumulative computational demand of a task (e.g., cumulative cognitive load). Accordingly, we use the framework as a qualitative lens to guide experimental design and interpretation, rather than as a quantitative metric of model's processing capacity.

**Cognitive Bandwidth.** We first define Cognitive Bandwidth ($B$) as a fundamental, latent property of a given pre-trained LLM ($\mathcal{M}$). It represents the model's total capacity to hold and manipulate information, maintain causal chains, and execute instructions within a single, coherent operational context. We posit that $B(\mathcal{M})$ is a fixed quantity for a particular LLM.

**Cognitive Load.** Next, we define Cognitive Load ($L$) as the computational demand a specific stage of a task imposes on the LLM. To analyze this, we need to decompose an long-horizon agent's workflow into discrete stages. We identify two primary workflows:

- **Action-Based Agent:** An iterative loop of **Environment Understanding (EU)** → Planning (PL) → Decision Making (DM).
- **Schema-Based Agent:** A sequence of EU → PL → **Schema Instantiation (SI)** → DM.

The most cognitive expensive stages are bolded respectively for different paradigms. All the stages will contribute to the total cognitive load and the cumulative cognitive load ($L_C$) is the sum of the loads of all stages in the workflow which could be expressed as $L_C = \sum L_{\text{stage}}$.

**Shifting the Cognitive Burden.** The central insight of our perspective lies in *how different agents based on distinct LLMs distribute this cognitive load*.

For **action-based agents**, the load is heavily concentrated in the **Environment Understanding (EU)** stage. Faced with a long, noisy context and an exhaustive list of possible actions, the agent must solve a "needle-in-a-haystack" problem, which severely impairs subsequent reasoning.

Conversely, **schema-based agents** transfer this burden to the **Schema Instantiation (SI)** stage. While this stage demands sophisticated semantic reasoning to ground abstract schemas into valid, executable actions, it benefits from a cleaner, more structured context provided by clean schema list.

Crucially, we posit that the load of a specific stage ($L_{\text{stage}}$) is malleable. While an agent's total bandwidth ($B$) is fixed, targeted training on tool-use or planning tasks can reduce the cognitive load of corresponding stages, thereby increasing the agent's overall task-completion capacity.

**The Failure Condition.** Putting it all together, the perspective predicts a task failure when the cumulative cognitive load surpasses the model's cognitive bandwidth:

$$\text{Failure} \iff L_C = \sum L_{\text{stage}} > B(\mathcal{M})$$

This framework allows us to conceptualize agent failure not as a random event, but as a predictable outcome of the interaction between a model's intrinsic limits and a task's architectural demands.

# 4 EXPERIMENTS AND ANALYSIS

This section presents our experimental setup, reports results across four environments spanning a range of action space complexities, where we observe a representation-choice inflection point and conducts a behavioral efficiency analysis across multiple LLMs to characterize the behavior differences between the two action representations.

## 4.1 EXPERIMENT SETUP

**Environments.** Our evaluation testbed comprises TextCraft (Prasad et al., 2024), WebShop (Yao et al., 2022), ALFWorld (Shridhar et al., 2021), and SciWorld (Wang et al., 2022). This selection is motivated by the progressively increasing action space complexity across these environments, as measured by the growing size of their grounded action lists and the number of action schemas.

**Evaluation Paradigm.** Following the methodology of Yuan et al. (2025), we evaluate agent performance on 100 tasks sampled from each environment. At each decision-making step, the agent is provided with the complete interaction history for reference and planning. The action information, either the grounded action list or schemas, is appended to the environment observation, but only when the set of available actions differs from the previous step, thus ensuring context efficiency. Based on preliminary observations that trajectories rarely succeed beyond 30 steps, we manually set this as the maximum interaction round for any given episode. For our evaluation metric, we adhere to the standard metrics established in the original benchmark papers: success rate for ALFWorld and average reward for all other tasks. Details of environments and interaction protocols are presented in Appendix B and information regarding selected models are provided in Appendix C.

## 4.2 IDENTIFYING THE REPRESENTATION PERFORMANCE INFLECTION POINT

Our cross-environmental analysis, presented in table 2, substantiates the principle that the selection of the action representation ought to be conditioned on the action space complexity of the task environment. Particularly, we identify a clear, non-monotonic performance curve as we traverse the complexity spectrum. In environments of low-to-moderate complexity (e.g. WebShop, ALFWorld), where the action space remains manageable, explicitly enumerating executable actions is superior. This approach bypasses the cognitive overhead of schema instantiation and benefit from a clean and concise environment observation, directly boosting performance. However, this advantage erodes and ultimately reverses in the high-complexity regime of SciWorld. Here, the combinatorial explosion of possible actions makes an exhaustive action list not only computationally expensive but also detrimental to performance. The action space size and context history can cumulatively exceed the model's context window and introduce overwhelming distracting noise. In this scenario, the abstraction afforded by a concise set of action schemas becomes essential, enabling the agent to reason effectively over an otherwise intractable decision space. **This trade-off pinpoints a critical representation-choice inflection point in task action space size, situated between ALFWorld and SciWorld. Below this choice inflection point, planning with actions is optimal; above it, planning with schemas is paramount for effective and scalable reasoning.**

## 4.3 PARADIGM BEHAVIORAL COMPARISON VIA EXPLORATION EFFICIENCY

To understand the underlying reasons for the performance shift at the inflection point, we dissect agent behavior by quantifying the agent exploration efficiency, following Zhang et al. (2025). we

Table 2: Model performance on environments of different action-space complexity.

| Model | TextCraft | | | WebShop | | | ALFWorld | | | SciWorld | | |
|---|---|---|---|---|---|---|---|---|---|---|---|---|
| | Actions | Schema | Delta | Actions | Schema | Delta | Actions | Schema | Delta | Actions | Schema | Delta |
| *Large Language Models* | | | | | | | | | | | | |
| Qwen2.5-7B | - | 26.0 | - | 22.0 | 28.5 | 6.5 | 49.0 | 27.0 | -22.0 | 8.0 | 8.2 | 0.2 |
| Qwen3-235B-A22B | - | 95.0 | - | 23.0 | 27.4 | 4.4 | 58.0 | 5.0 | -53.0 | 43.3 | 62.2 | 18.9 |
| Llama-4-Scout | - | 65.0 | - | 17.2 | 23.3 | 6.1 | 27.0 | 12.0 | -15.0 | 26.9 | 41.4 | 14.5 |
| DeepSeek-V3 | - | 59.0 | - | 24.7 | 13.3 | -11.4 | 50.0 | 7.0 | -43.0 | 22.0 | 25.5 | 3.5 |
| Kimi-K2 | - | 83.0 | - | 31.6 | 27.2 | -4.4 | 69.0 | 35.0 | -34.0 | 40.7 | 52.9 | 12.2 |
| Minimax-01 | - | 15.0 | - | 13.2 | 10.0 | -3.2 | 29.0 | 6.0 | -23.0 | 19.3 | 28.7 | 9.4 |
| GPT-4.1-mini | - | 96.0 | - | 20.3 | 14.6 | -5.7 | 44.0 | 12.0 | -32.0 | 46.0 | 44.8 | -1.2 |
| GPT-4.1 | - | 100.0 | - | 27.1 | 23.9 | -3.2 | 61.0 | 16.0 | -45.0 | 43.3 | 62.2 | 18.9 |
| Gemini-2.0-flash | - | 76.0 | - | 26.8 | 17.3 | -9.5 | 40.0 | 1.0 | -39.0 | 34.5 | 38.9 | 4.4 |
| *Large Reasoning Models* | | | | | | | | | | | | |
| DeepSeek-R1 | - | 97.0 | - | 36.7 | 36.0 | -0.7 | 65.0 | 14.0 | -51.0 | 48.4 | 46.9 | -1.5 |
| Seed-OSS-36B | - | 96.0 | - | 2.0 | 0.0 | -2.0 | 48.0 | 25.0 | -23.0 | 49.5 | 67.4 | 17.9 |
| Qwen3-235B-A22B$_{Thinking}$ | - | 96.0 | - | 39.7 | 20.0 | -19.7 | 47.0 | 23.0 | -24.0 | 47.7 | 67.3 | 19.6 |
| LongCat | - | 94.0 | - | 20.3 | 27.6 | 7.3 | 61.0 | 41.0 | -20.0 | 41.1 | 47.8 | 6.7 |
| GPT5-mini | - | 88.0 | - | 11.7 | 24.7 | 13.0 | 19.0 | 0.0 | -19.0 | 36.0 | 20.5 | -15.5 |
| Gemini-2.5-Flash | - | 88.0 | - | 18.0 | 15.0 | -3.0 | 40.0 | 8.0 | -32.0 | 57.2 | 69.1 | 11.9 |
| Claude-4.0-Sonnet | - | 98.0 | - | 64.0 | 64.0 | 0.0 | 87.0 | 27.0 | -60.0 | 60.2 | 69.1 | 8.9 |

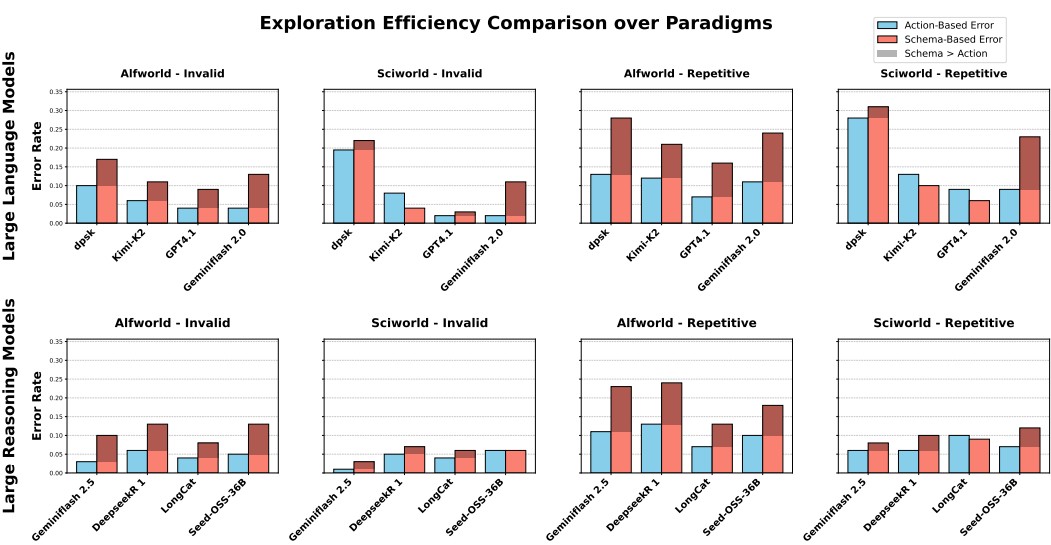

Figure 2: Comparison of agent behaviors in ALFWorld and SciWorld, two environments situated on either side of the observed complexity inflection point.

quantify two critical failure modes: (1) **Invalid Actions**, where an agent generates syntactically malformed commands, and (2) **Repetitive Actions**, defined as redundant steps that fail to advance the task. We systematically compare the planning with actions and planning with schemas paradigms by measuring the prevalence of these failures in agent trajectories across the ALFWorld and SciWorld environments over various models. The results reveal a clear dynamic that explains the performance trade-off. In the moderate-complexity environment of ALFWorld, the planning with actions paradigm is unequivocally superior, yielding significantly lower invalid and repetitive action rates. This suggests that when the action space is manageable, the cognitive cost of schema instantiation ($L_{SI}$) is an unnecessary burden that leads to frequent reasoning errors and wasteful exploration.

Conversely, in the high-complexity setting of SciWorld, this dynamic is inverted. Here, the cognitive burden shifts from schema instantiation to processing an overwhelmingly large grounded action list

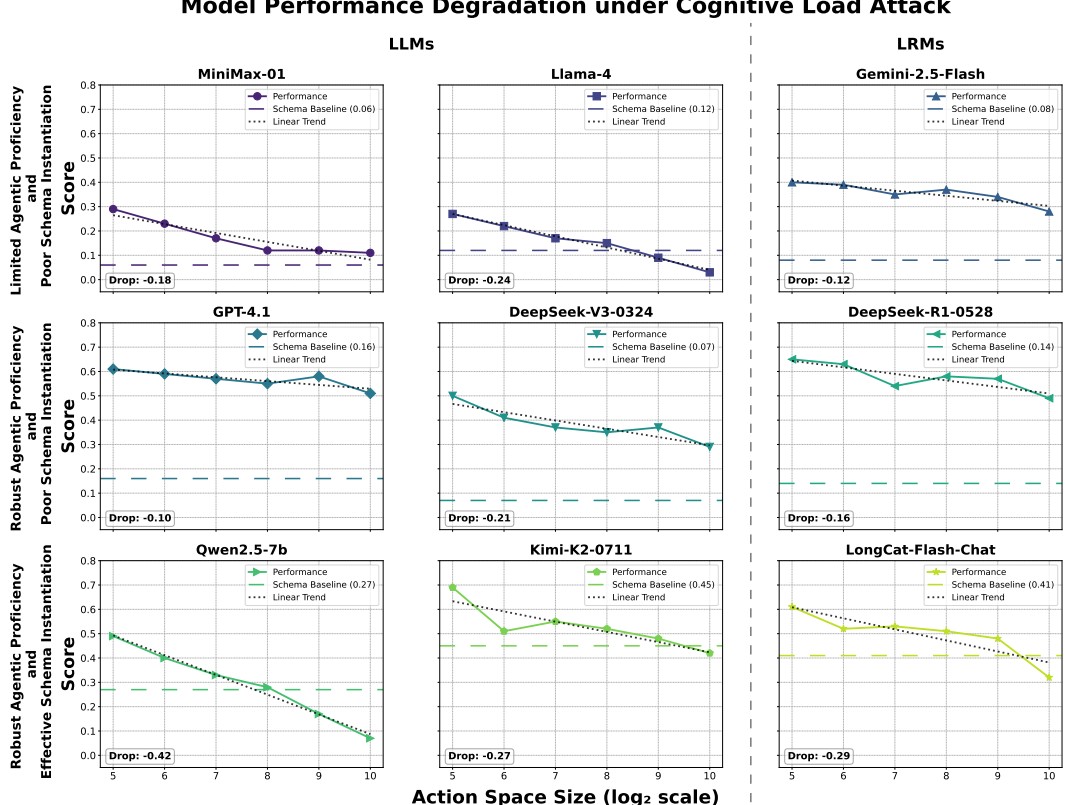

Figure 3: Effect of action-space size and environment complexity on success rate. In low-complexity settings (e.g., baseline ALFWorld), planning with actions (PwA) outperforms planning with schemas (PwS). As the action list expands with distractors, PwA performance declines and can be overtaken by PwS for sufficiently capable models, revealing the representation inflection point.

($L_{EU}$). For most models, the schema-based paradigm becomes more effective. This finding demonstrates that schemas provide a superior mechanism for managing extreme combinatorial complexity. By focusing the agent's reasoning on a tractable set of 26 templates instead of an intractable list of hundreds of actions, schemas enable more efficient decision-making, corroborating the transfer from planning with actions to planning with schemas when the environmental complexity scales. The trajectory analysis is done via rule-based filtering and Kimi-K2, details are shown in Appendix D.

# 5 DETERMINANTS OF THE REPRESENTATION INFLECTION POINT

To probe the dynamic of this inflection point regarding different model capabilities, we meticulously design cognitive load stress test experiments over various LLMs by increasingly inject invalid actions into executable action list to escalate the EU load imposed on LLMs to qualitatively discover how model capabilities influence the position of the inflection point.We conduct the experiment in ALFWorld, the environment close to the action space complexity of the inflection point.

Starting from the native action space (approximately 32 valid actions), we progressively augment the candidate list by injecting synthetically generated distractor actions until the prompt contains 1,024 actions in total. Distractors are syntactically plausible yet non-executable or task-irrelevant, so they cannot complete the task. This preserves task semantics and ensures that any performance degradation arises from screening a longer, noisier action list rather than from altered task structure. Noisy action examples are provided in Appendix E. We intentionally exceed SciWorld's grounded action-list length ($\sim$500 actions) because SciWorld's actions are largely executable and semantically similar, hence intrinsically more confusable, whereas our distractors, by design, are easier to dismiss. Using only 500 distractors would therefore under-approximate the EU burden; increasing the list to

1,024 compensates for their lower confusability and better approximates the effective EU load of high-complexity settings. Although distractors should in principle be filtered out, our hypothesis is that the sheer volume of candidates induces a "needle-in-a-haystack" burden on EU.

As shown in Figure 3, success rate declines monotonically as the action list grows. This trend is consistent with the Cognitive Bandwidth Perspective: enlarging the action space increases EU load, which in turn degrades PwA performance even when distractors are non-executable. The result strengthens the interpretation that the representation-choice inflection point could be driven by EU overload in PwA rather than by changes in task semantics.

However, beyond performance degradation of individual models, model's performance gap between paradigm differences exhibit different dynamics, where some models still perform superior with exhaustive actions while others underperform schema-based agents when the action space size is huge. Based on our observation, we manually categorize models across two axes. (i) Model's agentic proficiency—represented by it's performance under PwA. Specifically, a model without noisy actions perform better than 50% success rate in ALFWorld is considered as proficient. (ii) Model's schema instantiation capability—models under PwS score above 25% could be considered as effective, considering most of the models exhibit limited schema induction ability. According to our classification, contemporary models could be categorized in to three kinds, representing different mechanisms of model capability driving the representation-choice inflection point leftward or rightward.

**Category 1: Limited agentic proficiency and poor schema instantiation capability.** Models in this category, exhibit markedly low performance on ALFWorld under both the action-based and schema-based paradigms. Their performance also demonstrates a lack of robustness when subjected to cognitive load test. This dual failure indicates a fundamental deficiency in capabilities required for long-horizon tasks. These models struggle with the entire agentic pipeline, from understanding the environment and formulating a plan to inducing a task schema and making decisions. Consequently, every stage of a long-horizon task imposes a substantial cognitive burden. The failure condition is met as the $L_C$ from all cognitive components overwhelms the model's capacity as follows:

$$L_C = L_{\text{EU}} + L_{\text{PL}} + L_{\text{SI}} + L_{\text{DM}} > B(\mathcal{M})$$

With this deficency in handling long-horizon tasks and consequently high cognitive load by each stage of the task, **these models rarely benefit from PwS; the inflection point shifts rightward, making PwA the preferable representation across a wider complexity range.**

**Category 2: Robust agentic proficiency and poor schema instantiation capability.** This category includes models that demonstrate strong performance in the action-based paradigm but fail significantly in the schema-based paradigm. Even under the most intensive cognitive load test, their action-based performance remains superior, highlighting their robust agentic proficiency for long-horizon planning. The significant disparity in performance between the two paradigms points to a fragile or underdeveloped schema instantiation capability. For these models, navigating and understanding fuzzy or complex environments does not impose a prohibitive cognitive load. However, the process of identifying interactable objects, combining them with approriate schemas, and instantiating that schema into an executable actions becomes an insurmountable challenge. This focused bottleneck leads to failure, where the cognitive load is dominated by the schema-related processes:

$$L_C \approx L_{\text{SI}} + L_{\text{DM}} > B(\mathcal{M})$$

**Given robust agentic proficiency but poor schema instantiation capability, the inflection point also shifts rightward.** An explicit executable action list remains optimal up to higher complexities.

**Category 3: Robust agentic proficiency and effective schema instantiation capability.** Models in this final category possess both robust agentic decision-making abilities and effective schema induction mechanisms. They achieve strong performance in the action-based paradigm and demonstrate notable efficacy when planning with schemas. A key characteristic of these models is their response to escalating cognitive load. As the cognitive load attack intensifies, the performance of the action-based agents gradually degrades, to the point where the schema-based agents may outperform them. This suggests that the cognitive load associated with schema instantiation is comparable to that of understanding a noisy environment from raw inputs. These models can handle either load most of the time, indicating a mature and balanced set of agentic capabilities. Task failure is only met occasionally and is not systemic. The failure condition is therefore met under specific circumstances

where one of these components becomes particularly demanding:

$$L_C \approx L_{\text{SI}} + L_{\text{DM}} \text{ or } L_{\text{EU}} + L_{\text{DM}} > B(\mathcal{M})$$

**With balanced capabilities, the inflection point shifts leftward: PwS becomes favorable earlier as complexity scales for these family of models.**

## 6 WHAT MAKES CAPABLE AGENTS PLANNING WITH SCHEMA

With the inflection point tied to model capability and PwA limited by intractably long action lists, we focus on improving PwS to shift the inflection left and make schema-based planning broadly viable. Below, we outline empirically supported arguments that could serve as a actionable insight towards building more capable schema-based agents.

**Long Reasoning: Beneficial, but Not Critical** While models with enhanced long-reasoning capabilities generally exhibit improved task performance, this advantage does not directly translate to a greater aptitude for schema instantiation. Consequently, when the task mandates planning with schemas, the performance gains are marginal. This finding suggests that although a greater reasoning depth could expand an LLM's cognitive bandwidth, this enhancement is insufficient to overcome the core deficiency. If the fundamental capability for schema instantiation is lacking, this bottleneck will persist and lead to task failure. Thus, **we conclude that long-reasoning capability, while generally advantageous, is not the decisive factor in mastering PwS.**

**The key recipe: post-training for multi-turn tool use** We argue that the training methodology of an LLM plays the critical role. Specifically, a specialized post-training paradigm can reduce the cognitive load imposed by schema instantiation, enabling the model to operate effectively when planning with schemas. For LLMs like Kimi-K2 (Bai et al., 2025) and LongCat (MeituanLongCatTeam et al., 2025), training that incorporates heavy agentic reinforcement learning with multi-turn tool-use data appears to distinguish them from other contemporary models. Particularly, Kimi-K2 manually collects more than 3000 MCP tools from web repositories and synthetically generate more than 20000 tools across diverse domains. Following that, a lot of agents are assigned with different set of tools to generate task and trajectories of different difficulties for further tool-use post training needs. Similar training procedures are also found in the technical report of Longcat. Their outstanding performance on complicated tool-use benchmarks like $\tau$-bench (Yao et al., 2024), $\tau^2$-bench (Barres et al., 2025), and ACEBench (Chen et al., 2025) further validate this. Since these multi-turn tool-use resources require the model to fill in particular parameters for a tool to be executed to obtain further results. We posit that this skill, learned from populating structured tool calls, can be seamlessly transferred to the long-horizon task of schema instantiation, thereby reducing the cognitive load required at this stage. Hence, **we contend that enhancing LLMs through post-training with multi-turn, tool-use-focused agentic training is a fundamental and impactful direction for the future development of agents under PwS.**

## 7 CONCLUSION

We systematically investigated the optimal action representation for long-horizon agents as environmental action space scales, focusing on two action representations, PwA and PwS. To understand the differences between conventional PwA and our proposed alternative PwS, we introduce the Cognitive Bandwidth Perspective, which decomposes agent workflows into distinct stages. Framed as a qualitative lens rather than a measurable construct, this perspective helps explain when and why PwS scales better. Empirically, across four environments of increasing action-space size, we observe an inflection point between ALFWorld ($\sim$35 actions) and SciWorld ($\sim$500 actions). Below this regime, PwA is preferable because SI overhead dominates; above it, PwS is superior as the EU burden of parsing long, noisy action lists becomes prohibitive, indicating that the optimal representation is contingent on scale and that PwS offers better asymptotic scalability, becoming better action representation for real world autonomy. We perform cognitive-load stress test by injecting distractors into ALFWorld to probe mechanism and observe that the inflection location shifts with two capability axes. Only models with effective SI shift the inflection leftward, making PwS viable over a broader complexity range. Based on our experiment results and the training technique of various models, we provide actionable insights for shifting the inflection leftward and enables scalable agents: post-training that emphasizes multi-turn tool could be essential.

ETHICS STATEMENT

We affirm our commitment to the ICLR Code of Ethics. Our research does not involve human subjects or the collection of new personally identifiable information. We uses only public, text-based simulators (TextCraft, WebShop, ALFWorld, SciWorld) and off-the-shelf models for inference-only evaluation. All the prompts used in this study are appended in the appendix and no harmful content is involved. We emphasize that the Cognitive Bandwidth Perspective is a qualitative lens and simulator results may not transfer to open-world deployment; any real-world use requires additional alignment, monitoring, and access control.

REPRODUCIBILITY STATEMENT

All LLM evaluations in this work were conducted via public APIs, specifically using the Open-Router platforms, with the total experimental cost estimated at 5,000 USD. Details regarding model selection are illustrated in Appendix C. We provide all prompt templates and system implementations in Appendix to facilitate full reproducibility of our experiments. All code and data will be publicly released to foster future research.

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

APPENDIX

# A    ACTION SCHEMA LISTS FOR DIFFERENT ENVIRONMENTS

## A.1    ALFWORLD

**Action Schema List for ALFWorld**

```
Alfworld_Template = [
``close [OBJ]'',
``cool [OBJ] with [OBJ]'',
``heat [OBJ] with [OBJ]'',
``examine [OBJ]'',
``go to [OBJ]'',
``inventory'',
``look'',
``open [OBJ]'',
``put [OBJ] in/on [OBJ]'',
``take [OBJ] from [OBJ]'',
``clean [OBJ] with [OBJ]'',
]
```

## A.2    SCIWORLD

**Action Schema List for SciWorld**

```
Sciworld_Template = [
``open [OBJ]'',
``close [OBJ]'',
``activate [OBJ]'',
``deactivate [OBJ]'',
``connect [OBJ] to [OBJ]'',
``disconnect [OBJ]'',
``use [OBJ]'',
``use [OBJ] on [OBJ]'',
``look around'',
``look at [OBJ]'',
``look in [OBJ]'',
``read [OBJ]'',
``move [OBJ] to [OBJ]'',
``pick up [OBJ]'',
``put down [OBJ]'',
``pour [OBJ] into [OBJ]'',
``dunk [OBJ] into [OBJ]'',
``mix [OBJ]'',
``go to [OBJ]'',
``eat [OBJ]'',
``flush [OBJ]'',
``focus on [OBJ]'',
``wait'',
``wait1'',
``task'',
``inventory''
]
```

# B    EVALUATION ENVIRONMENTS AND EXPERIMENT PROTOCOLS

Our experiments are conducted on a suite of four environments, selected to represent a progression of increasing complexity. The environments are modified from AgentGym (Xi et al., 2025).

**Textcraft** (Prasad et al., 2024): TextCraft is a text-only simulation environment designed to evaluate an agent's capacity for compositional reasoning and planning. Agents are tasked with crafting a specific target item by issuing text commands, with tasks requiring a variable number of steps to complete. The action space is restricted to three commands: craft, get, and inventory, forcing the agent to learn the hierarchical structure of crafting recipes.

**Webshop** (Yao et al., 2022): WebShop is an interactive, simulated e-commerce environment designed to evaluate an agent's ability to act on compositional, natural language instructions. Given

a detailed product description, the agent must navigate the website, search for products, and select an item that satisfies as many specified attributes as possible. The agent's action space consists of searching with free-form text and clicking buttons on the webpage. A final reward is calculated based on the degree of match between the purchased item and the user's initial instruction

**ALFWorld** (Shridhar et al., 2021): ALFWorld is a simulated household environment designed to benchmark an agent's ability to execute high-level tasks. Success in this environment requires the agent to perform multi-step reasoning, navigate and explore various rooms, and interact with objects. The action space supports a range of commands, including object manipulation and navigation. The environment provides deterministic feedback on the outcome of each action based on a predefined logical framework that governs the world state.

**SciWorld** (Wang et al., 2022): ScienceWorld is an interactive text-based benchmark created to evaluate an agent's scientific reasoning capabilities. The environment features a diverse set of 30 task types derived from a standard elementary science curriculum, such as correctly using measurement instruments or conducting simple mechanics experiments. Following each action executed by the agent, the environment simulator provides an updated observation, describing the effects and changes within the simulated world.

**Experiment Protocols.** The prompts for interacting with environment API are fully aligned with AgentGym (Xi et al., 2025) with minimal modification from self-deployed models to model API calling.

## C   MODEL SELECTION

Our study conducts analyses across two distinct classes of models: general-purpose large language models and specialized large reasoning models. For the LLM category, we select a suite of state-of-the-art models representing a spectrum of scales and architectures: Qwen2.5-7B (Yang et al., 2024), DeepSeek-V3 (DeepSeek-AI et al., 2024), Kimi-K2 (Bai et al., 2025), GPT-4.1, Claude and so on. This selection is curated to capture performance trends across models with varying parameter counts. For the LRM category, we chose DeepSeek-R1 (DeepSeek-AI et al., 2025) and Longcat (MeituanLongCatTeam et al., 2025), as they are prominent examples of recent models specifically engineered for complex tasks.

Full list of selected models are provided below.

---

**Large Language Models without Long Reasoning Capability**

```
Opensourced:  Qwen2.5-7B (Yang et al., 2024)
Qwen3-235B-A22B-NonThinking (Yang et al., 2025)
Llama4-Scout
DeepSeek-V3 (DeepSeek-AI et al., 2024)
Kimi-K2 (Bai et al., 2025)
Minimax-01 (MiniMax et al., 2025)
Proprietary
GPT-4.1-mini
GPT-4.1
Gemini-2.0-flash (Anil et al., 2023)
```

---

**Large Reasoning Models**

```
Opensourced:  DeepSeek-R1 (DeepSeek-AI et al., 2025)
Seed-OSS-36B
Qwen3-235B-A22B-Thinking
LongCat (MeituanLongCatTeam et al., 2025)
Proprietary
GPT5-mini
Gemini-2.5-Flash
Claude-4.0-Sonnet
```

---

# D    TRAJECTORY ANALYSIS PROMPT

## D.1    ACTION ANALYSIS FOR REPETITIVE ACTIONS

This analysis is conducted based on the analysis result of Kimi-K2, a powerful LLMs with 1T parameters effectively trained for agentic tasks.

---

**System Prompt For Action Analysis**

```
ACTION_ANALYSIS_SYS_PROMPT = You are an expert trajectory analyst.
Your task is to identify repetitive/meaningless actions in an agent's interaction history.

**Input:**
- A trajectory dictionary mapping integer step indices to interaction strings.
Each interaction string has the format:
  "Human: <observation>\n Agent: <thought and action>"

**Definition of Repetitive Action:**
- **Meaningless repeated actions** that don't advance the task
- Must meet these criteria:
  1. The action is identical or functionally equivalent to a previous action
  2. The action occurs after the environment has provided negative feedback
  3. The action doesn't yield new information or state change
  4. The action doesn't contribute to task progression
- Examples:
  - Repeatedly opening the same empty drawer after being told it's empty
  - Moving between the same locations without purpose
  - Checking inventory multiple times without new items
  - Retrying an action immediately after failure feedback

**Task:**
- Analyze the trajectory and identify step indices for repetitive actions
- Output a JSON dictionary with exactly one key:
  - "repetitive_action": sorted list of step indices with repetitive actions

**Output Rules:**
- Only include step indices where the action clearly meets the criteria
- Sort indices in ascending order
- If no instances are found, return an empty list
- Output must be valid JSON with no additional text

**Output Schema:**
{
  "repetitive_action": [<int>, <int>, ...]
}

**Analysis Guidelines:**
1. Focus on patterns:
   - Look for repeated action patterns, not necessarily consecutive
   - Consider actions that are functionally equivalent
   (e.g., "open cabinet 1" and "open cabinet1")
   - Pay attention to environment feedback (e.g., "nothing happens", "still empty")

2. Be practical:
   - Include actions that show clear lack of progress
   - Ignore repetitions that are part of necessary sequences
   - Consider the context of the task

3. Key indicators:
   - Environment feedback indicating no change
   - Agent thoughts showing frustration or confusion
   - Actions that could be optimized (e.g., moving between same locations)

**Example:**
Input:
{
  "0": "Human: Kitchen with cabinet\n Agent: Open cabinet",
  "1": "Human: Cabinet is empty\n Agent: Close cabinet",
  "2": "Human: Cabinet closed\n Agent: Open cabinet",
  "3": "Human: Cabinet still empty\n Agent: Open cabinet",
  "4": "Human: Cabinet remains empty\n Agent: Look for key",
  "5": "Human: Key on counter\n Agent: Take key",
  "6": "Human: Key taken\n Agent: Open cabinet with key",
  "7": "Human: Cabinet opens\n Agent: Take microwave",
  "8": "Human: Microwave taken\n Agent: Put microwave in fridge",
  "9": "Human: Microwave doesn't fit\n Agent: Put microwave in fridge",
  "10": "Human: Still doesn't fit\n Agent: Try putting microwave in fridge"
```

```
    }

    Output:
    {
      "repetitive_action": [2, 3, 9, 10]
    }

    Explanation:
    - Step 2: Repetitive (reopening cabinet after knowing it's empty)
    - Step 3: Repetitive (consecutive identical action after known empty)
    - Step 9: Repetitive (retrying after failure feedback)
    - Step 10: Repetitive (functionally equivalent retry after failure)
    - Step 6: Not repetitive (new context with key)
```

**System Prompt For Action Analysis**

```
    ACTION_ANALYSIS_USER_PROMPT = Analyze the following trajectory
    and identify invalid actions and repetitive actions.

    Trajectory:
    {trajectory}

    Return JSON output exactly as specified.
```

## D.2  ACTION ANALYSIS FOR INVALID ACTIONS

The analysis for invalid actions is conducted via on rule-based filtering. Specifically, we count invalid action frequency by detecting the signature response from the environment API which indicates invalid. For example, "nothing happens." for API response in ALFWorld and "No known action matches that input."

## E  NOISY ACTION LIST FOR COGNITIVE LOAD ATTACK

In this section, we demonstrate some schema-aligned simplified examples on the distractors to be injected into ALFWorld action list.

**Invalid OBJ Noisy Actions**

```
    ALFWorld_Distractors = [
        "open portal",
        "open mirror",
        "open book",
        "open map",
        "open pyramid",
        "open record",
        "open portrait",
        "open vehicle",
        "open council",
        "open grove",
        "open ship",
        "open library",
        "close portal",
        "close mirror",
        "close book",
        "close map",
        "close pyramid",
        "close record",
        "close portrait",
        "close vehicle",
        "close council",
        "examine feather",
        "examine crystal",
        "examine mane",
        "examine stone",
        "examine sensor",
        "examine rune",
        "examine balloon",
        "examine charm",
        "examine star",
```

```
        "examine wing",
        "examine mirror",
        "examine book",
        "examine map",
        "examine record",
        "examine portrait",
        "examine pyramid",
        "examine vehicle",
        "examine council",
        "examine grove",
        "examine library",
        "examine ship",
        "examine algorithm",

        # go to
        "go to grove",
        "go to portal",
        "go to plane",
        "go to realm",
        "go to world",
        "go to border",
        "go to library",
        "go to garden",
        "go to council",
        "go to ocean",

        # put [OBJ] in/on [OBJ]
        "put feather in pyramid",
        "put crystal in library",
        "put book on portrait",
        "put map on council",
        "put charm in vehicle",
        "put mirror on pyramid",
        "put stone in grove",
        "put mane on portrait",
        "put star on garden",
        "put wing on council",
        "put record in library",
        "put form on portrait",
        "put song in library",
        "put algorithm in library",
        "put tree in garden",

        # take [OBJ] from [OBJ]
        "take book from library",
        "take map from council",
        "take feather from pyramid",
        "take crystal from garden",
        "take charm from vehicle",
        "take mirror from portrait",
        "take stone from grove",
        "take wing from council",
        "take record from library",
        "take star from world",

        # clean [OBJ] with [OBJ]
        "clean mirror with feather",
        "clean pyramid with feather",
        "clean portrait with mane",
        "clean book with charm",
        "clean record with feather",
        "clean council with cloth",
        "clean library with cloth",
        "clean vehicle with feather",
        "clean grove with cloth",

        # heat [OBJ] with [OBJ]
        "heat book with incense",
        "heat crystal with incense",
        "heat stone with incense",
        "heat mirror with incense",
        "heat record with incense",
        "heat pyramid with incense",

        # cool [OBJ] with [OBJ]
        "cool crystal with wave",
        "cool book with radiation",
        "cool mirror with wave",
        "cool stone with radiation",
```

```
        "cool pyramid with wave",
        "cool portrait with radiation",

        # extra plain variants
        "open council",
        "close council",
        "examine message",
        "examine pattern",
        "examine aura",
        "examine vision",
        "examine orb",
        "examine fabric",
        "examine machine",
        "examine tree",
        "examine path",
        "examine portrait",
        "examine riddle",
        "examine math",
        "examine wave",
        "go to territory",
        "go to library",
        "go to realm",
        "put book in library",
        "put map in library",
        "put song in library",
        "put form in library",
        "put algorithm on council",
        "put feather on portrait",
        "put crystal on council",
        "take charm from portrait",
        "take balloon from garden",
        "take stone from council",
        "take vision from library",
        "take song from library",
        "clean portrait with feather",
        "clean book with mane",
        "clean pyramid with charm",
    ]
```

# F  LONG HORIZON TASK FORMULATION

Formally, an agent in an interactive environment could be modeled as decision-making under partial observation. Following existing works (Yuan et al., 2025; Song et al., 2024), these tasks could be formulated as **Partially Observable Markov Decision Process (POMDP)** which could be expressed in form of $(\mathcal{U}, \mathcal{S}, \mathcal{A}, \mathcal{O}, \mathcal{T}, \mathcal{R})$. To be detailed, the $\mathcal{U}$ stands for the task description and the relevant requirements. $\mathcal{S}$ is the state space, $\mathcal{A}$ represents the action space of the agent, and $\mathcal{O}$ is the observation space. $\mathcal{T}$ stands for the transition function where $\mathcal{T} : \mathcal{S} \times \mathcal{A} \rightarrow \mathcal{S}$ and $\mathcal{T}$ is determined by the interactive environment. $\mathcal{R}$ represents the reward function $\mathcal{R} : \mathcal{S} \times \mathcal{A} \rightarrow [0, 1]$ which indicates the final reward of the agents movements. Since our experiments are natural language based, $\mathcal{U}, \mathcal{S}, \mathcal{A}$ and $\mathcal{O}$ are all in natural language form. At each timestamp t, the trajectory history is denoted as:

$$\tau_t = (u, a_1, o_1, \ldots, a_t, o_t) \sim \pi_\theta(\tau_t|u) \tag{1}$$

where $a_i \in \mathcal{A}, o_i \in \mathcal{O}$ stand for the action and the observation after the action $a_t$ is executed at timestamp t. The probability of agent with parameter $\theta$ generating $\tau$ would be:

$$\pi_\theta(\tau|u) = \prod_{j=1}^{n} \pi_\theta(a_j|i, a_1, o_1, \ldots, o_{j-1}) \tag{2}$$

where n is the trajectory length. Finally, a final reward r is computed, with 1 indicating success of the task, 0 stands for the failure of the task and other values for partial completion of the task.

# G  THE USE OF LARGE LANGUAGE MODELS

In this paper, the LLMs serves as a writing assistant to help polish the content.

