# OpenReview forum: "The Cognitive Bandwidth Bottleneck: Shifting Long-Horizon Agent from Planning with Actions to Planning with Schemas"
_ICLR.cc/2026/Conference — Submitted to ICLR 2026_

### Official Review · Reviewer_J1fi · 2025-10-30

**Soundness:** 2
**Presentation:** 2
**Contribution:** 2
**Rating:** 2
**Confidence:** 3

**Summary:**

This paper proposes a cognitive perspective on how LLM agents handle planning in environments of increasing complexity. The authors argue that action-based agents suffer from cognitive overload as the action space expands, whereas schema-based agents alleviate this burden by reasoning over abstract action schemas. The paper introduces the cognitive bandwidth perspective which models the agent’s reasoning as a limited resource distributed over stages such as environment understanding, planning, and schema instantiation. Experiments across four text-based environments, TextCraft, WebShop, ALFWorld, and SciWorld, show that planning with action performs better in simple settings while planning with schema scales better to larger complicated environments. The study positions schema-based reasoning as a cognitively efficient alternative for long-horizon agent scalability.

**Strengths:**

* Clear conceptual motivation: The paper tackles a fundamental but often overlooked question of how agents should represent actions. This framing provides a fresh angle for studying reasoning efficiency in LLM-based agents for long-horizon tasks.
* Cognitive theory integration: The idea of modeling LLM reasoning through cognitive bandwidth is novel and offers an intuitive explanation for performance drops in complex environments.
* Systematic environment scaling: Using four progressively complex benchmarks from TextCraft to SciWorld gives a logical structure to the experimental comparison.
* Controlled cognitive stress test: The authors cleverly simulate increasing environment understanding load by injecting distractor actions, demonstrating how planning with action performance collapses under noise while planning with schema remains stable.

**Weaknesses:**

* Weak rationale for the cognitive-load claims: The claim that planning with actions' load concentrates in environment understanding and planning with schemas shifts it to schema instantiation is never empirically validated. No token-level or attention-based measurement of reasoning load is provided. The argument relies only on indirect correlations e.g., error rates rather than direct evidence. For example, in an environment where abundant number of different action schemas exist, this assumption cannot always hold.
* Inaccurate characterization of environment complexity: The paper states that ALFWorld has ~35 actions, but ALFWorld which is based on ALFRED actually contains hundreds of grounded action combinations from 11 action schemas and 84 object classes. The authors should provide a detailed explanation of how this figure of 35 actions was derived.
* Unclear explanation of ALFWorld results: Planning with schemas performance in ALFWorld is extremely low (e.g., GPT-4.1 = 16 %, Gemini-2.0-flash = 1%), despite the schema space being small (11 schemas with 35 actions). I think such failure cannot be explained just by SI overhead. An in-depth analysis and a plausible explanation are needed.
* Lack of methodological transparency and reproducibility: The paper’s Reproducibility Statement promises all prompt templates and implementations in the appendix, but none are provided. Key details like prompt format, environment wrappers, model parameters, and handling of invalid actions are missing. This limits proper comprehension and appreciation of the provided results as well as the reproducibility of the results.
* Presentation clarity issues (tables and figures): In Table 2, the “Delta” column lacks an explicit definition that makes readers infer that it means (PwS – PwA) difference. Also in Figure 2, the legend for the gray shading incurs confusion as the grey colored bar does not exist in the chart. As such, all the figures and tables need to be reexamined and revised for clearer presentation.

**Questions:**

Please refer to the Weaknesses section for the raised issues.

---

> ### Author Response · Authors · 2025-11-22
>
> Dear Reviewer cDAS:
>
> Thanks for your review and your acknowledgment on our work in the following areas:
> - Clear conceptual motivation for studying optimal action representations.
> - The idea of modeling LLM reasoning through cognitive bandwidth is novel and offers an intuitive explanation
> - Using four progressively complex benchmarks from TextCraft to SciWorld gives a logical structure to the experimental comparison.
> - Cleverly simulate increasing environment understanding load by injecting distractor actions, demonstrating how planning with action performance collapses under noise.
>
> We believe the following paragraphs could address the weakness mentioned by the reviewer.
>
> > Weakness 1: Weak rationale for the cognitive-load claims: The claim that planning with actions' load concentrates in environment understanding and planning with schemas shifts it to schema instantiation is never empirically validated. No token-level or attention-based measurement of reasoning load is provided. The argument relies only on indirect correlations e.g., error rates rather than direct evidence.
>
>
> We thank the reviewer for pointing out that a more direct piece of evidence for the cognitive load shift was missing in the original paper. To address this concern, we provide a comparison of the **environment understanding load (EU load)**—approximated by token counts—for PwA and PwS on **ALFWorld** and **SciWorld**.
>
> Here, *progress* is defined as the ratio of current step over total steps in a trajectory.
>
> **Kimi-K2 token count on ALFWorld**
>
> | Progress (Current_Step/Total_Steps) | PwS Failure Trail | PwS Success Trail | PwA Failure Trail | PwA Success Trail |
> |:-----------------------------------:|:-----------------:|:-----------------:|:-----------------:|:-----------------:|
> | 0.0 | 289  | 289  | 289  | 289  |
> | 0.2 | 821  | 750  | 1618 | 1070 |
> | 0.4 | 1128 | 1013 | 2845 | 1506 |
> | 0.6 | 1634 | 1216 | 3997 | 2099 |
> | 0.8 | 2259 | 1443 | 4184 | 2597 |
> | 1.0 | 2751 | 1853 | 4505 | 3171 |
>
> **Kimi-K2 token count on SciWorld**
>
> | Progress (Current_Step/Total_Steps) | PwS Failure Trail | PwS Success Trail | PwA Failure Trail | PwA Success Trail |
> |:-----------------------------------:|:-----------------:|:-----------------:|:-----------------:|:-----------------:|
> | 0.0 | 501   | 501   | 501    | 501    |
> | 0.2 | 1650  | 1174  | 10962  | 3603   |
> | 0.4 | 2364  | 2036  | 30766  | 6561   |
> | 0.6 | 3500  | 2536  | 50604  | 16956  |
> | 0.8 | 5008  | 3474  | 79293  | 27254  |
> | 1.0 | 6019  | 4234  | 111707 | 45560  |
>
> From these results, we observe:
>
> - As **environment complexity increases** (ALFWorld → SciWorld), the **EU load under PwA grows explosively**, especially in the later stages of the trajectory.
> - Under **PwS**, the EU load remains in a **much more moderate range**, even in SciWorld and even for failure trajectories.
>
> This provides **direct quantitative evidence** for our claim that, in complex environments, **PwA suffers from a severe explosion in long‑context understanding load**, while **PwS keeps this component of cognitive load more controlled**.
>
> ---
>
> ### Rationale for the cognitive load shift
>
> The rationale for a cognitive load shift from **environment understanding (EU)** to **schema instantiation (SI)** is twofold:
>
> 1. **Token-based EU load evidence**
>
>    The token statistics above show that, as complexity grows, **PwA’s EU load increases dramatically**, whereas **PwS keeps EU load comparatively moderate**. This supports the view that **PwS mitigates long‑context understanding pressure** relative to PwA in the *same* environment.
>
> 2. **Performance differences and procedural asymmetry**
>
>    At the same time, there is a clear **performance difference** between PwA and PwS (two sides of the “inflection point” discussed in the paper), which cannot be explained by EU load alone.
>
>    As illustrated in Figure 1 of our paper, the **structural difference** between PwA and PwS lies in how executable actions are obtained:
>
>    - Under **PwA**, actions are **directly provided by the environment**.
>    - Under **PwS**, actions must be **internally produced via schema instantiation**.
>
>    This PwS pipeline requires the model to:
>
>    1. Recognize available objects,
>    2. Understand action schemas,
>    3. Instantiate schemas with objects, and
>    4. Make a final decision over the instantiated actions **within a single response**.
>
>    Since this is the main procedural difference between PwA and PwS, we attribute the **additional difficulty (and performance drop) under PwS** to a **shift in cognitive load** from long‑context understanding to **schema instantiation and grounding**.
>
> ---
>
> Taken together, the **token‑based EU load analysis** and the **observed performance patterns** both support our central claim:
>
> > **PwS reduces long‑context understanding pressure but introduces a substantial cognitive burden on schema instantiation, which current models are not yet good at handling.**

---

> > ### Author Response · Authors · 2025-11-22
> >
> > > Weakness 1: For example, in an environment where abundant number of different action schemas exist, this assumption cannot always hold.
> >
> > We thank the reviewer for raising this subtle point about environments with many action schemas and their impact on cognitive load.
> >
> > Our view can be summarized as follows:
> >
> > **1. Executable actions vs. schemas**
> >
> > In environments with an abundant number of different action schemas, the size of the **executable action list** can explode, as we empirically observe when comparing ALFWorld vs. SciWorld. Schemas act as **abstract representations of families of executable actions**. Consequently, for a **fixed environment**, the number of schemas is generally **much smaller** than the number of executable actions that would need to be enumerated under PwA.
> >
> > **2. Interpreting “many schemas”**
> >
> > We understand the reviewer’s concern as follows: if an environment itself contains a **large number of distinct schemas**, then the **context‑understanding load under PwS** might still be very high. We agree that as the number of schemas increases, the environment‑understanding pressure for PwS also increases.
> >
> > However, our claim about a **“load shift”** is always intended **within a fixed environment**, i.e., holding the underlying environment constant and only changing the interaction paradigm (PwA vs. PwS). Under this assumption:
> >
> > - When many schemas exist, **PwS must reason more** about which schema to instantiate and how.
> > - But if the same environment is run under **PwA**, the agent is additionally confronted with a **much larger list of fully instantiated executable actions** in the context. This amplifies the **long‑context understanding burden** and exacerbates a “needle‑in‑a‑haystack” problem over actions.
> >
> > **3. Why our assumption still holds**
> >
> > Therefore, even though a **schema‑rich environment** can raise environment‑understanding pressure in absolute terms, **PwA amplifies this pressure further** by surfacing a **combinatorially larger action list** in the prompt. By contrast:
> >
> > - **PwS** keeps the context more compact (**schemas + observations**),
> > - and shifts part of the burden to **schema instantiation and grounding**.
> >
> > Under a **fixed environment**, we thus still believe that:
> >
> > - **PwS mitigates long‑context understanding pressure relative to PwA**,
> > - while **transferring cognitive load** to schema instantiation and grounding.
> >
> > We hope this clarifies that our “cognitive load shift” assumption is always meant as a **within‑environment comparison**, and under this lens, the argument for **PwS as a more scalable paradigm** still generally holds.

---

> > > ### Author Response · Authors · 2025-11-22
> > >
> > > > Weakness 2: Inaccurate characterization of environment complexity: The paper states that ALFWorld has ~35 actions, but ALFWorld which is based on ALFRED actually contains hundreds of grounded action combinations from 11 action schemas and 84 object classes. The authors should provide a detailed explanation of how this figure of 35 actions was derived.
> > >
> > > We thank the reviewer for pointing this out and acknowledge that our explanation in the paper was not sufficiently clear.
> > >
> > > While ALFWorld is indeed a complex environment with a very large **global action space**, our analysis focuses on the **effective action space that the LLM must choose from at each interaction step**. In our 100-sample subset used for experiments, the **average size of the available action list per turn is only about 35 actions**.
> > >
> > > Our notion of **environment complexity** is therefore **not** based on the total number of possible actions in the environment, but on the **per-step decision space** that the agent actually faces during interaction. From this perspective, **ALFWorld is substantially less complex than environments like SciWorld**, where the per-step action choice is much larger, and this is the axis of complexity relevant to our scalability analysis.

---

> > > > ### Author Response · Authors · 2025-11-22
> > > >
> > > > > Weakness 3: Unclear explanation of ALFWorld results: Planning with schema’s performance in ALFWorld is extremely low (e.g., GPT-4.1 = 16 %, Gemini-2.0-flash = 1%), despite the schema space being small (11 schemas with 35 actions). I think such failure cannot be explained just by SI overhead. An in-depth analysis and a plausible explanation are needed.
> > > >
> > > > We thank the reviewer for bringing this point up.
> > > >
> > > > We view this analysis as a natural extension of the behavioral study in Section 4.3. From the figure, we observe that the rates of **repetitive actions** and **invalid actions** are substantially higher when the agent operates under the **PwS** paradigm. In particular, PwS agents tend to suffer from a **self-error conditioning** problem: once an error is made, the agent becomes overly focused on “fixing” that local mistake, gets trapped in a local minimum, and loses sight of overall task progress.
> > > >
> > > > This trapped behavior causes many steps to be wasted and, after repeated failed attempts, the agent often effectively abandons meaningful progress, leading to task failure. We believe that **limited zero-shot tool-use capability** is a key underlying factor: it (i) increases the likelihood of invalid actions and (ii) reduces the agent’s ability to recover once it has gone off track, thereby amplifying these repetitive and self-conditioned failure patterns under PwS.

---

> ### Author Response · Authors · 2025-11-22
>
> > Weakness 4: Lack of methodological transparency and reproducibility
>
> We apologize for the lack of clarity in our description of the experimental methodology. Our implementation is based on the open‑source agent framework AgentGym. Below, we provide additional details on our experimental setup to make our methodology transparent and reproducible.
>
> **Interaction Prompts**
>
> For all environments, an initial round of interaction is provided as a system prompt detailing the task requirements. The LLM then acknowledges with:
>
> > OK. I'll follow your instructions and try my best to solve the task.
>
> ---
>
> **TextCraft**
>
> System prompt:
>
> > You are given few useful crafting recipes to craft items in Minecraft. Crafting commands are of the format "craft [target object] using [input ingredients]".
> > Every round I will give you an observation, you have to respond an action based on the state and instruction. You can "get" an object (ingredients) from the inventory or the environment, look-up the game inventory by "inventory", or "craft" [0-9] (target) using any of the crafting commands, for example, "get 4 stone". You can use ONLY these crafting commands provided, do not use your own crafting commands. However, if the crafting command uses a generic ingredient like "planks", you can use special types of the same ingredient e.g. "dark oak planks" in the command instead.
> > Your response should use the following format:
> >
> > Thought:
> > ...
> >
> > Action:
> > ...
>
> ---
>
> **WebShop**
>
> System prompt:
>
> > You are web shopping.
> > I will give you instructions about what to do.
> > You have to follow the instructions.
> > Every round I will give you an observation and a list of available actions, you have to respond an action based on the state and instruction.
> > You can use search action if search is available.
> > You can click one of the buttons in clickables.
> > An action should be of the following structure:
> > `search[keywords]`
> > `click[value]`
> > If the action is not valid, perform nothing.
> > Keywords in search are up to you, but the value in click must be a value in the list of available actions.
> > Remember that your keywords in search should be carefully designed.
> > Your response should use the following format:
> >
> > Thought:
> > I think ...
> >
> > Action:
> > click[something]. Notice that nothing else should be included in the Action part.
>
> ---
>
> **ALFWorld**
>
> System prompt:
>
> > Interact with a household to solve a task. Imagine you are an intelligent agent in a household environment and your target is to perform actions to complete the task goal. At the beginning of your interactions, you will be given the detailed description of the current environment and your goal to accomplish. For each of your turn, you will be given a list of actions which you can choose one to perform in this turn. You should choose from two actions: "THOUGHT" or "ACTION". If you choose "THOUGHT", you should first think about the current condition and plan for your future actions, and then output your action in this turn. Your output must strictly follow this format:
> > "Thought:
> > your thoughts.
> >
> > Action:
> > your next action"
> >
> > If you choose "ACTION", you should directly output the action in this turn. Your output must strictly follow this format:
> > "Action:
> > your next action".
> >
> > After your each turn, the environment will give you immediate feedback based on which you plan your next few steps. if the envrionment output "Nothing happened", that means the previous action is invalid and you should try more options.
> >
> > Reminder:
> > 1. the action must be chosen from the given available actions. Any actions except provided available actions will be regarded as illegal.
> > 2. Think when necessary, try to act directly more in the process.
>
> ---

---

> > ### Author Response · Authors · 2025-11-22
> >
> > > Weakness 4 (Cont.): Lack of methodological transparency and reproducibility
> >
> > **SciWorld**
> >
> > System prompt:
> >
> > > You are an agent for science world. Every round I will give you an observation, you have to respond an action based on the observation to finish the given task. Here are the actions you may take:
> > > [{"action": "open/close OBJ", "description": "open/close a container"}, {"action": "de/activate OBJ", "description": "activate/deactivate a device"}, {"action": "connect OBJ to OBJ", "description": "connect electrical components"}, {"action": "disconnect OBJ", "description": "disconnect electrical components"}, {"action": "use OBJ [on OBJ]", "description": "use a device/item"}, {"action": "look around", "description": "describe the current room"}, {"action": "look at OBJ", "description": "describe an object in detail"}, {"action": "look in OBJ", "description": "describe a container's contents"}, {"action": "read OBJ", "description": "read a note or book"}, {"action": "move OBJ to OBJ", "description": "move an object to a container"}, {"action": "pick up OBJ", "description": "move an object to the inventory"}, {"action": "put down OBJ", "description": "drop an inventory item"}, {"action": "pour OBJ into OBJ", "description": "pour a liquid into a container"}, {"action": "dunk OBJ into OBJ", "description": "dunk a container into a liquid"}, {"action": "mix OBJ", "description": "chemically mix a container"}, {"action": "go to LOC", "description": "move to a new location"}, {"action": "eat OBJ", "description": "eat a food"}, {"action": "flush OBJ", "description": "flush a toilet"}, {"action": "focus on OBJ", "description": "signal intent on a task object"}, {"action": "wait", "description": "take no action for 10 iterations"}, {"action": "wait1", "description": "take no action for 1 iteration"}, {"action": "task", "description": "describe current task"}, {"action": "inventory", "description": "list your inventory"}]
> > > Your response should use the following format:
> > >
> > > Thought:
> > > your thoughts.
> > >
> > > Action:
> > > your next action. Notice the generated action shouldn't be with special format like **** or other special formats.
> >
> > **Model Parameters**
> >
> > For the open-sourced models adopted in our experiments, we list the model details below:
> >
> > - **Llama4-Scout-109B-A17B**
> > - **Qwen2.5-7B**
> > - **Qwen3-235B-A22B**
> > - **DeepSeek-V3-671B-A37B**
> > - **DeepSeek-R1-671B-A37B**
> > - **Kimi-K2-1T-A32B**
> > - **LongCat-Thinking-560B-27B** (the activated parameteres are reported in average)
> > - **Minimax-01-456B-A45.9B**
> > - **Seed-OSS-36B**
> >
> > ---
> >
> > **Environment Wrappers and Handling of Invalid Actions**
> >
> > We implement all environment wrappers **following the AgentGym implementation**, and **do not modify** its default behavior.
> >
> > - When **invalid actions** occur, the environment **is not terminated**.
> > - Some environments return **special feedback**:
> >   - ALFWorld: `"Nothing happened"`
> >   - SciWorld: `"No known action matches that input"`
> > - Other environments continue as normal **without special signals**.
> >
> > In all cases, the environment feedback (including any invalid-action messages) is **concatenated to the interaction history** and provided back to the agent for subsequent decisions.
> >
> > ---
> >
> > **Method of Providing Actions in the Prompt**
> >
> > We provide **actions and action schemas** in the prompt **only when the set of executable actions changes**, i.e., when the environment undergoes a **significant change** relative to previous interactions. This strategy reduces redundant prompt content while ensuring that the agent is updated whenever the action space meaningfully shifts.

---

> ### Author Response · Authors · 2025-11-22
>
> > Weakness 5: Presentation clarity issues
>
> We thank the reviewer for pointing out that the figure captions were unclear. We shortened the captions to stay within the page limit and we will clarify them in the revision for disambiguity.
>
> In **Figure 2**, “Delta” denotes the performance gap **PwS − PwA**. We also use color coding to indicate which paradigm performs better:
> - **Green**: PwS outperforms PwA
> - **Red**: PwA outperforms PwS
>
> Regarding the **gray legend** in Figure 2, it corresponds to the **shaded regions** that highlight the differences between the bars. The shading is intended to make the performance gaps more visually salient.
>
> In the revised version, we will (i) explicitly explain this in the caption and (ii) adjust the visualization to make the legend–shading correspondence more immediately understandable.
>
> We appreciate the reviewer’s suggestion and will improve the figure design and captions accordingly.
>
> ***We sincerely appreciate your valuable advice and hope that our response will assist you in raising your score. Thank you once again!***

---

> > ### Comment · Reviewer_J1fi · 2025-11-25
> > **Thank you for your clarifying responses**
> >
> > The responses address my concerns, so I am raising my score to 6. I hope the revised version will include the clarifications and additional quantitative results.

---

> > > ### Author Response · Authors · 2025-11-25
> > >
> > > Dear Reviewer J1fi,
> > >
> > >
> > > Thank you for your response, and we are happy to see that our response address your concerns. Thank you again!
> > >
> > >
> > > Sincerely,
> > >
> > > Authors of Submission 7312

---

### Official Review · Reviewer_cDAS · 2025-10-31

**Soundness:** 4
**Presentation:** 4
**Contribution:** 4
**Rating:** 6
**Confidence:** 4

**Summary:**

This paper explores the "Cognitive Bandwidth Bottleneck" faced by large language models (LLMs) when handling long-horizon tasks and compares two action representation paradigms: **Planning with Actions (PwA)** and **Planning with Schemas (PwS)**. The cognitive bandwidth bottleneck occurs when LLMs are unable to effectively process long-term task planning due to limited information processing capacity. PwA, which uses predefined action sequences, works well for short tasks but is inefficient for long-term tasks. In contrast, PwS, which employs schemas to represent task components, handles long-term dependencies more effectively, offering greater flexibility and improved task planning. The paper concludes that PwS provides a better solution for long-horizon tasks by overcoming the cognitive bandwidth limitation of PwA.

**Strengths:**

## Strengths

The paper introduces a novel framework for comparing two action representation paradigms—**Planning with Actions (PwA)** and **Planning with Schemas (PwS)**—through the lens of **Cognitive Bandwidth**. This conceptual framework is particularly effective in explaining how the cognitive capacity of large language models (LLMs) is allocated during task execution. By formalizing the trade-offs between PwA and PwS, the framework highlights that PwS offers a more scalable and flexible approach, inspired by human cognition, by reducing the complexity of action representations through schemas. In contrast, PwA relies on a detailed action list, which, while effective for short tasks, is less efficient for long-horizon tasks due to the high cognitive load at the environment understanding (EU) stage. PwS, by shifting the cognitive load to the schema instantiation (SI) stage, allows for better handling of long-term dependencies, making it a more reliable and effective choice for complex tasks. This insight into cognitive bandwidth and its impact on action representation provides a strong theoretical foundation for advancing LLM-based task planning systems.

**Weaknesses:**

## Weaknesses

While the paper provides a solid theoretical framework for comparing **Planning with Actions (PwA)** and **Planning with Schemas (PwS)**, several weaknesses in the experimental evaluation and analysis can be identified:

1. **Limited Experimental Scope for PwA**: No results are provided  in TextCraft on PwA scenarios. So the results are not clear in simple environments. I recommend the **babyAI** benchmark could serve as a needed test for PwA, which might give more evidence of your contribution.

2. **Insufficient Comparison in Complex Scenarios**: Although **AlfWorld** is expected to be a more complex environment than **WebShop**, the experimental results show that **PwS** does not significantly outperform PwA in this case. This discrepancy raises questions about whether PwS is truly more effective in handling more intricate tasks, as the framework suggests. Further clarification are needed to explain this result and whether additional factors such as task characteristics or environmental constraints influence performance.

3. **Confuse about Relationship between Action Paradigm Choice and Environment**: The paper also needs to provide a discussion on how the choice between PwA and PwS might relate to the environment itself. Specifically, it is unclear whether the selection of the action representation paradigm is solely dependent on the intrinsic properties of large language models (LLMs) or if it also interacts with environmental factors and low-level skills. A more detailed explanation of how the environment, task complexity, and model capabilities influence the choice between PwA and PwS would provide more practical insight into their applicability in real-world scenarios.

**Questions:**

See weakness above.

---

> ### Author Response · Authors · 2025-11-22
>
> Dear Reviewer cDAS:
>
> Thanks for your review and your acknowledgment on our work in the following areas:
> - Propose novel framework in comparing two different action representations.
> - Conceptually demonstrate and formalize the differences between action representations.
> - The insight into cognitive bandwidth and its impact on action representation provides a strong theoretical foundation for LLM-based task planning systems.
>
> We believe the following paragraphs could address the weakness mentioned by the reviewer.
>
> > Weakness 1: Limited Experimental Scope for PwA:
>
> We are delighted by the reviewer’s insightful comment that BabyAI could be a more suitable environment for our experiments.
>
> To address this concern, we ran BabyAI under both PwA and PwS setups on 4 representative models: 3 non-thinking models from three different categories (as classified in our paper) and 1 long‑reasoning model (LRM). The results are:
>
> | Model           | PwA   | PwS   | Δ (PwS − PwA) |
> |-----------------|:-----:|:-----:|:-------------:|
> | DeepSeek-V3     | 17.1  | 6.2   |    −10.9      |
> | Llama-4-Scout   | 31.1  | 20.8  |    −10.3      |
> | Kimi-K2         | 47.2  | 30.9  |    −16.3      |
> | DeepSeek-R1     | 47.8  | 25.3  |    −22.5      |
>
> From these results, we observe that:
>
> - **PwA consistently outperforms PwS** on BabyAI across all evaluated models.
> - This is consistent with our main conclusion that **current LLMs still fall short under the PwS paradigm**, especially in relatively simple environments where the per‑step action space is manageable.
>
> In simpler environments like BabyAI, **PwA remains the more effective practical choice** for today’s models. However, we still maintain that **PwS is the more scalable paradigm in principle**, particularly as environment complexity and action combinatorics grow. We view the present results as evidence that PwS is currently under-optimized and under-studied.
>
> Additionally, we would also like to clarify our findings on **TextCraft (TC)**.
>
> In TextCraft, we observe that models already achieve satisfying performance under the **PwS** paradigm, in contrast to their behavior in ALFWorld and BabyAI. We view this as evidence that TC is a **comparatively simple environment**, where the gap between PwA and PwS is intrinsically small.
>
> Placing this in the broader picture of **environment complexity**, our empirical results across benchmarks suggest the following qualitative relationship between environment complexity and the relative performance of PwA vs. PwS:
>
> - **Very simple environments (e.g., TextCraft):**
>   PwA and PwS behave similarly, and models can perform well even under PwS.
>
> - **Moderate‑complexity environments (e.g., ALFWorld):**
>   PwA tends to outperform PwS, as current models still struggle with robust schema instantiation and grounding.
>
> - **High‑complexity environments (e.g., SciWorld):**
>   PwS starts to show its advantages in terms of scalability (e.g., token efficiency and reduced long‑context burden), even though today’s models are not yet fully capable of exploiting this potential.
>
> Thus, the TextCraft results are **consistent with our overall claim**: in simple environments, PwA and PwS can be comparable, but as complexity increases, **PwS is expected to become the more scalable paradigm**, provided that models are better trained for schema instantiation and environment sensing.

---

> > ### Author Response · Authors · 2025-11-22
> >
> > > Weakness 2: Insufficient Comparison in Complex Scenarios. Although AlfWorld is expected to be a more complex environment than WebShop, the experimental results show that PwS does not significantly outperform PwA in this case.
> >
> > We appreciate the reviewer for giving us the opportunity to clarify this point.
> >
> > From our perspective, the **inflection point observed between ALFWorld and SciWorld** is the key evidence that PwS constitutes a more scalable paradigm. The gap in environment complexity between these two benchmarks—e.g., in terms of the number of available actions and templates—is **substantially larger** than that between WebShop and ALFWorld, suggesting that the latter pair does not differ enough in scale to induce a dramatic performance shift.
> >
> > In contrast, the relatively strong PwS performance in **WebShop** can plausibly be attributed to **task and interface characteristics** rather than pure complexity. WebShop is rendered in a web-like format that closely matches the distributions modern LLMs are heavily trained on, making the environment more “natural” for current models under the PwS paradigm. This web-related familiarity may explain why PwS performs comparatively well in WebShop, whereas the clearer scalability signal comes from the **ALFWorld–SciWorld transition**.
> >
> > Taken together, these observations still support our framework: **PwS reveals its advantages as environment complexity truly scales up**. At the same time, we agree that additional factors such as task characteristics and environmental constraints also influence performance, as illustrated by the behavior on WebShop.

---

> > > ### Author Response · Authors · 2025-11-22
> > >
> > > > Weakness 3: Confuse about Relationship between Action Paradigm Choice and Environment
> > >
> > > We thank the reviewer for raising this important point.
> > >
> > > First, we would like to clarify the **motivation for proposing the PwS paradigm**. Our primary consideration is **real‑world applications**, where PwA is often unrealistic: making a decision based on a *comprehensive, externally provided executable action list* is typically impractical. For example, for a household robot, it is hard to imagine an API that, after every movement, returns the full list of executable actions available in the environment. Instead, such a robot must **maintain and instantiate its own action templates** based on perception and prior knowledge—precisely the setting captured by **PwS**. In this sense, we view PwS as the more *practical* and *realistic* paradigm for real-world decision making.
> > >
> > > Second, we summarize the **relationship between environment complexity and the choice of action paradigm**:
> > >
> > > - For a **fixed model with fixed capabilities**, in environments with **higher complexity than SciWorld** (e.g., substantially larger action spaces or many more interactable objects), we would **recommend using PwS**. In our experiments, almost all models demonstrate relative advantages of PwS in such complex settings.
> > > - For **lower‑complexity environments**, such as **ALFWorld**, PwA is still generally **superior in practice** with current models, and thus we would recommend PwA as the ideal paradigm in these benchmarks.
> > > - When considering **additional factors**, such as environments with particular formats (e.g., **web-like interfaces**), adopting PwS in relatively easier environments can also be reasonable. Models may perform comparatively well under PwS in these cases because they have been trained on similar data distributions or interface formats.
> > >
> > > To sum up, while **PwA remains preferable in today’s low‑complexity benchmarks**, **PwS better reflects real‑world constraints** (where no oracle action list is available) and is **more promising as a scalable long‑term paradigm** as environment complexity continues to grow. **A general guidance could follow our model classification established in Section 5.**
> > >
> > > ***We sincerely appreciate your valuable advice and hope that our response will assist you in raising your score. Thank you once again!***

---

> > > > ### Author Response · Authors · 2025-11-25
> > > >
> > > > Dear Reviewer cDAS,
> > > >
> > > > As the rebuttal period is drawing to a close, we would be very grateful for any further comments or suggestions you might have on our submission. We hope that our additional experiments and clarifications have addressed your concerns and contributed positively to your assessment of the paper.
> > > >
> > > > Sincerely,
> > > >
> > > > The authors of submission 7312

---

### Official Review · Reviewer_4W3y · 2025-10-31

**Soundness:** 3
**Presentation:** 3
**Contribution:** 2
**Rating:** 4
**Confidence:** 3

**Summary:**

This paper addresses the scaling challenge for Large Language Model (LLM) agents in environments with combinatorially exploding action spaces by comparing two core action representations: Planning with Actions (PwA) and Planning with Schemas (PwS). PwS aims to ensure scalable and compliant action generation by instantiating an abstract schema (e.g., "move [OBJ] to [OBJ]") into a concrete action (e.g., "move apple to desk").

Empirical results across environments of varying complexity (ALFWorld $\approx$ 35 actions to SciWorld $\approx$ 500 actions) reveal a representation-choice inflection point. PwA is superior in the low-to-medium action space due to lower schema instantiation (SI) overhead, but PwS gains an advantage in high-complexity environments (SciWorld) where the environment understanding load of PwA becomes prohibitive. This suggests the optimal representation is task-dependent. The paper concludes that post-training focused on multi-turn tool use enhances SI capability, shifting the inflection point leftward.

**Strengths:**

1. The PwS paradigm is promising for real-world autonomy as it aligns with human cognitive mechanisms and inherently produces interface-compliant outputs that scale more reliably than free-form generation.
2. The conclusions offer actionable steps for future agent development, specifically identifying that post-training which emphasizes multi-turn tool use significantly improves Schema Instantiation (SI) capability, which is necessary to shift the inflection point to the left and make PwS broadly viable.
3. Empirical evidence is compelling, demonstrating a Representation-Choice Inflection Point located between ALFWorld ($\approx$ 35 actions) and SciWorld ($\approx$ 500 actions), providing quantified evidence of the scaling limitations inherent in the conventional PwA paradigm.
4. I like Figure 1 which shows a high-level overview of the relationship between environment complexity and agent effectiveness. Helped me understand the paper better.

**Weaknesses:**

1. The Cognitive Bandwidth Perspective is explicitly defined as a conceptual and qualitative framework, lacking any quantifiable metrics or probes for the postulated bandwidth ($\mathcal{B}(\mathcal{M})$) or load ($\mathcal{L}_{stage}$) variables.

2. The paper notes that PwS agents demonstrate suboptimal performance compared to PwA in the low-to-medium complexity ALFWorld, suggesting the fundamental bottleneck of Schema Instantiation (SI) is underdeveloped in current models, despite being the proposed solution for scalability.

3. The behavioral comparison's analysis of "Repetitive Actions" and "Invalid Actions" relies on either simple rule-based filtering (for Invalid Actions) or an external LLM (Kimi-K2) to analyze and generate ground truth for failure modes, which could introduce potential noise and dependence on a specific model's subjective judgment.

**Questions:**

1. To overcome the qualitative nature of the Cognitive Bandwidth Perspective, could the authors propose a quantifiable proxy for Schema Instantiation (SI) load ($\mathcal{L}_{SI}$), perhaps based on the number of objects visible in the observation (combinatorial complexity of grounding) or the token count of the generated action string? This could help replace the subjective binary categorization in Section 5.
2. The paper argues PwA suffers from high Environment Understanding (EU) load and long context pressure. Yet, for PwS, the final grounded action still needs to be inserted into the history for subsequent interaction steps (Eq. 1, 2). Could the authors measure the total historical token growth for successful trajectories in SciWorld under both PwA and PwS? This would provide quantitative evidence on how much PwS actually mitigates the "long context pressure" throughout the entire multi-turn trajectory.
3. The current schemas for ALFWorld and SciWorld are derived by replacing all object arguments with a generic `[OBJ]` token. How robust is this simple abstraction across different instruction styles? Does providing the LLM with slightly different schema formats, such as `move [FRUIT] to [DESK]` (using explicit types) or `move [OBJ1] to [OBJ2]`, significantly change the inflection point's location for models categorized with poor SI capability?
4. Since Long Reasoning Models (LRMs) have marginally improved PwS performance but failed to solve the SI bottleneck, did the authors experiment with prompting strategies that specifically augment the SI stage? For example, using a Chain-of-Thought (CoT) prompt structured as: "Thought: To instantiate 'move [OBJ] to [OBJ]', I must first identify available objects, then check if they are movable, and finally generate the action 'move apple to desk'"? Measuring performance change here would directly probe the bottleneck.

Would appreciate answers to these questions. I am willing to increase my scores if the rebuttal is convincible for my questions.

---

> ### Author Response · Authors · 2025-11-22
>
> Dear Reviewer 4w3y:
>
> Thanks for your review and your acknowledgment on our work in the following areas:
> - The PwS paradigm is promising for real-world autonomy as it aligns with human cognitive mechanisms and inherently produces interface-compliant outputs that scale more reliably than free-form generation.
> - The conclusions offer actionable steps for future agent development.
> - Empirical evidence is compelling, providing quantified evidence of the scaling limitations inherent in the conventional PwA paradigm.
> - The figure is clearly plotted.
>
> We believe the following paragraphs could address the weakness mentioned by the reviewer.
>
> > Weakness 1 and Question 1: The Cognitive Bandwidth Perspective is explicitly defined as a conceptual and qualitative framework, lacking any quantifiable metrics. To overcome the qualitative nature of the Cognitive Bandwidth Perspective, could the authors propose a quantifiable proxy for Schema Instantiation.
>
> We sincerely appreciate the reviewer’s curiosity about formalizing our method with a quantitative measure. We are actively working in this direction, aiming to quantify the cognitive load of both the schema instantiation stage and the long‑context understanding stage.
>
> We find your suggestion—measuring the cognitive load of schema instantiation via the number of objects visible in the observation (i.e., the combinatorial complexity of grounding) or the token length of the generated action string—particularly insightful, and it closely aligns with our own thinking. We have begun developing a similar quantitative model and are pleased to share both our current progress and the limitations of this preliminary approach in this rebuttal.
>
> **Definition of our current model.**
>
> We decompose the cognitive load at time step $t$ as:
> $$
> P_{\text{cognitive}}(t) = P_{\text{action}}(t)\cdot P_{\text{long}}(t),
> $$
> where
> $$
> P_{\text{action}}(t) = P_{\text{conception}}(t)\cdot P_{\text{pruning}}(t)\cdot P_{\text{decision}}(t),
> $$
> and
> $$
> P_{\text{long}}(t) = f \cdot \Bigl(1 + \frac{T_N[t]}{T_S[t]}\Bigr).
> $$
>
> Here, $P_{\text{cognitive}}(t)$ denotes the overall cognitive load the model faces at time $t$.
>
> - $P_{\text{action}}(t)$ captures the load of decision making, decomposed into:
>   - $P_{\text{conception}}(t)$: load of *conceiving* possible actions,
>   - $P_{\text{pruning}}(t)$: load of *pruning* these actions into a feasible set,
>   - $P_{\text{decision}}(t)$: load of *selecting* one action from the executable list.
>
> - $P_{\text{long}}(t)$ captures the *long‑context understanding* pressure. The scalar $f$ is a hyperparameter; $T_N[t]$ is the token count of “noisy” rounds that do **not** contribute to task progress, and $T_S[t]$ is the token count of “informative” rounds that **do** contribute. Informative vs. noisy rounds are automatically classified by Kimi‑K2, as manual annotation would be prohibitively expensive.
>
> We instantiate this model differently for PwA and PwS:
>
> - **PwA agents.** The environment directly provides an executable action list at each step, so we set
>   $$
>   P_{\text{conception}}(t) = 1,\quad P_{\text{pruning}}(t) = 1,
>   $$
>   and the action load is dominated by
>   $$
>   P_{\text{action}}(t) = P_{\text{decision}}(t),
>   $$
>   where $P_{\text{decision}}(t)$ reflects the pressure of choosing one action from the provided list (e.g., as a function of list length).
>
> - **PwS agents.** The model must instantiate actions from the observation:
>   - $P_{\text{conception}}(t)$ depends on how many interactable objects exist and, hence, how many candidate actions can be *imagined* from them (combinatorial grounding complexity).
>   - $P_{\text{pruning}}(t)$ measures the pressure of pruning these imagined actions into *practical executable actions*. When we fill interactable objects into schema placeholders, some invalid actions are also generated and must be filtered out by the model.
>   - $P_{\text{decision}}(t)$ is again the load of choosing one action from the final executable list.

---

> > ### Author Response · Authors · 2025-11-22
> >
> > Using this formulation, we approximate the cognitive pressure of Kimi‑K2 on ALFWorld and SciWorld, aggregated by trajectory progress (Current\_Step / Total\_Steps) and split by paradigm (PwA vs. PwS) and outcome (success vs. failure).
> >
> > **Kimi‑K2 cognitive pressure on ALFWorld**
> >
> > | Progress (Current\_Step / Total\_Steps) | PwS Failure Trail | PwS Success Trail | PwA Failure Trail | PwA Success Trail |
> > |:----------------------------------------|:-----------------:|:-----------------:|:-----------------:|:-----------------:|
> > | 0.0 | 1.0    | 1.0    | 1.0   | 1.0   |
> > | 0.2 | 2167.1 | 627.4  | 42.4  | 22.0  |
> > | 0.4 | 2401.3 | 1176.1 | 57.4  | 23.5  |
> > | 0.6 | 3478.4 | 1149.5 | 75.2  | 22.4  |
> > | 0.8 | 4808.9 | 1951.3 | 78.7  | 26.0  |
> > | 1.0 | 5856.3 | 1924.5 | 84.7  | 22.1  |
> >
> > **Kimi‑K2 cognitive pressure on SciWorld**
> >
> > | Progress (Current\_Step / Total\_Steps) | PwS Failure Trail | PwS Success Trail | PwA Failure Trail | PwA Success Trail |
> > |:----------------------------------------|:-----------------:|:-----------------:|:-----------------:|:-----------------:|
> > | 0.0 | 1.0      | 1.0       | 1.0    | 1.0   |
> > | 0.2 | 51540.9  | 39356.8   | 725.7  | 471.2 |
> > | 0.4 | 79802.0  | 48252.2   | 1825.3 | 476.6 |
> > | 0.6 | 197345.7 | 218398.4  | 1923.1 | 472.0 |
> > | 0.8 | 179410.2 | 95411.6   | 1927.2 | 473.1 |
> > | 1.0 | 182786.4 | 104877.0  | 1931.4 | 346.2 |
> >
> > From these results, we observe several qualitative trends:
> >
> > 1. The cognitive pressure in **SciWorld** is consistently higher than in **ALFWorld**, suggesting that SciWorld is a more cognitively demanding environment.
> > 2. The cognitive pressure under **PwS** is consistently higher than under **PwA** in both environments.
> > 3. Within each paradigm, the cognitive pressure of **failure** trajectories is generally higher than that of **success** trajectories.
> >
> > At the same time, our current model has important limitations:
> >
> > - The **scale of PwS and PwA is not aligned**. Because we model
> >   $$
> >   P_{\text{cognitive}}(t) = P_{\text{action}}(t)\cdot P_{\text{long}}(t)
> >   $$
> >   and define $P_{\text{long}}(t)$ via the ratio $\frac{T_N[t]}{T_S[t]}$, the overall cognitive pressure is numerically dominated by the schema‑instantiation / decision‑making term, especially in PwS. This makes the absolute values between PwS and PwA not directly comparable and indicates that better normalization or a different functional form (e.g., additive or re‑weighted) is needed.
> >
> > We view this as a first‑step approximation rather than a final metric. We are actively working on refining this framework into a more comprehensive and better‑calibrated measure of cognitive pressure, and we again thank the reviewer for encouraging this direction and would be happy to discuss alternative formulations.

---

> ### Author Response · Authors · 2025-11-22
>
> > Weakness 2: The paper notes that PwS agents demonstrate suboptimal performance compared to PwA in the low-to-medium complexity ALFWorld, suggesting the fundamental bottleneck of SI is underdeveloped in current models, despite being the proposed solution for scalability.
>
>
> We thank the reviewer for highlighting this important point.
>
> We agree that, in their current instantiations, PwS agents still underperform PwA agents in low‑to‑medium complexity environments such as ALFWorld. **Our aim in this work is not to claim that PwS is superior in all settings, but to propose a paradigm that is *more scalable* to substantially more complex environments.** This scalability potential is supported empirically by our experiments in SciWorld, where **we observe an inflection point at higher complexity levels: as task difficulty increases, PwS begins to outperform PwA.**
>
> We therefore **interpret the suboptimal PwS performance in simpler environments as a limitation of current LLMs’ schema‑instantiation capabilities, rather than as a fundamental flaw of the paradigm itself.** Bridging this gap—i.e., **improving LLMs’ SI capabilities so that PwS becomes competitive even in low‑to‑medium complexity environments**—is an important direction for future work. We see our results as clarifying the path from current agents to **truly scalable** ones.

---

> ### Author Response · Authors · 2025-11-22
>
> > Weakness 3: The behavioral comparison's analysis of "Repetitive Actions" and "Invalid Actions" relies on either simple rule-based filtering (for Invalid Actions) or an external LLM (Kimi-K2) to analyze and generate ground truth for failure modes, which could introduce potential noise and dependence on a specific model's subjective judgment.
>
> We thank the reviewer for raising this concern.
>
> For the analysis of **invalid actions**, we would like to clarify that the environment provides *explicit* feedback when an action is invalid (e.g., ALFWorld’s “Nothing happens” response), which allows us to identify such cases **deterministically**. As a result, our rule-based filtering does not rely on heuristic interpretation and does not introduce additional noise.
>
> For **repetitive actions**, the external LLM (Kimi-K2) is not asked to make open-ended or subjective judgments; instead, it is instructed to detect **explicit and well-defined patterns**, such as sequences of identical actions or consecutive invalid actions. This substantially reduces subjectivity compared to unconstrained labeling. We closely follow the implementation described by Zhang et al. [1], and we believe this setup is reliable and consistent with prior work.
>
> [1] Zijing Zhang, Ziyang Chen, Mingxiao Li, Zhaopeng Tu, and Xiaolong Li. RLVMR: reinforce
> ment learning with verifiable meta-reasoning rewards for robust long-horizon agents.

---

> ### Author Response · Authors · 2025-11-22
>
> > Question 2: The paper argues PwA suffers from high Environment Understanding (EU) load and long context pressure. Yet, for PwS, the final grounded action still needs to be inserted into the history for subsequent interaction steps
>
> We thank the reviewer for requesting additional evidence regarding EU cognitive load. To better illustrate the load shift between PwA and PwS, we report a progress-based token comparison on ALFWorld and SciWorld, where this effect is most clearly observable.
>
> Here, *progress* is defined as the ratio of current step over total steps in a trajectory.
>
> **Kimi-K2 Token Count on ALFWorld**
> | Progress (Current_Step/Total_Steps) | PwS Failure Trail | PwS Success Trail | PwA Failure Trail |  PwA Success Trail |
> |:-----------|:-------:|:-----------:| :-------:|:-----------:|
> | 0    |   289   |   289  |   289   |  289   |
> | 0.2 |  821  |  750  |  1618  |   1070 |
> | 0.4 |  1128  |  1013 |  2845  |  1506    |
> | 0.6 |  1634  |  1216 |  3997  |   2099   |
> | 0.8 |  2259  |  1443 |  4184  |  2597   |
> | 1.0 |  2751  |  1853 |  4505 |   3171   |
>
>
> **Kimi-K2 token count on SciWorld**
>
> | Progress (Current\_Step / Total\_Steps) | PwS Failure Trail | PwS Success Trail | PwA Failure Trail | PwA Success Trail |
> |:----------------------------------------|:-----------------:|:-----------------:|:-----------------:|:-----------------:|
> | 0.0 | 501   | 501   | 501    | 501    |
> | 0.2 | 1650  | 1174  | 10962  | 3603   |
> | 0.4 | 2364  | 2036  | 30766  | 6561   |
> | 0.6 | 3500  | 2536  | 50604  | 16956  |
> | 0.8 | 5008  | 3474  | 79293  | 27254  |
> | 1.0 | 6019  | 4234  | 111707 | 45560  |
>
> From this table, especially in SciWorld, we observe:
>
> - **PwS significantly reduces total tokens** compared to PwA across all progress intervals, for both success and failure trajectories.
> - **Token growth with trajectory length is much steeper in PwA than in PwS**. As episodes progress, the token counts under PwA explode, whereas PwS grows comparatively more moderately.
>
> We view this as complementary evidence that, in complex long-horizon environments such as SciWorld, **PwS can be more token-efficient and thus more scalable than PwA**, even though current models still underperform on PwS in terms of task success in simpler environments. This supports our core claim that PwS is a more promising long-term paradigm for scalable agents, while also highlighting that **better training and modeling for PwS is needed** to fully realize this potential.

---

> > ### Author Response · Authors · 2025-11-22
> >
> > > Question 3: The current schemas for ALFWorld and SciWorld are derived by replacing all object arguments with a generic [OBJ] token. How robust is this simple abstraction across different instruction styles?
> >
> > We thank the reviewer for raising the question of prompt robustness.
> >
> > To evaluate this, we conducted additional experiments on ALFWorld and SciWorld, where we modified the original prompt by changing the object and location placeholders (e.g., turning `[OBJ]` into `[OBJ1]` and `[OBJ2]`, and “location” into `[LOC]`). We then re-ran experiments on 3 non-thinking models (from different categories as classified in our paper) and 1 thinking model (DeepSeek-R1). The results are:
> >
> > **ALFWorld**
> >
> > | Model           | Prompt in Paper | Alternative Prompt |
> > |-----------------|:---------------:|:------------------:|
> > | DeepSeek-V3     | 7.0             | 5.0                |
> > | Llama-4-Scout   | 12.0            | 9.0                |
> > | Kimi-K2         | 45.0            | 51.0               |
> > | DeepSeek-R1     | 14.0            | 9.0                |
> >
> > **SciWorld**
> >
> > | Model           | Prompt in Paper | Alternative Prompt |
> > |-----------------|:---------------:|:------------------:|
> > | DeepSeek-V3     | 25.5            | 28.4               |
> > | Llama-4-Scout   | 41.4            | 32.2               |
> > | Kimi-K2         | 52.9            | 49.2               |
> > | DeepSeek-R1     | 46.9            | 43.6               |
> >
> > Overall, the performance differences between the original and alternative prompts are relatively small, and there is no systematic trend that would alter our main conclusions. While some models fluctuate slightly up or down, the qualitative patterns reported in the paper remain unchanged. We therefore believe the results are **reasonably robust** to this class of prompt variations.

---

> > > ### Author Response · Authors · 2025-11-22
> > >
> > > > Question 4: Since Long Reasoning Models (LRMs) have marginally improved PwS performance but failed to solve the SI bottleneck, did the authors experiment with prompting strategies that specifically augment the SI stage?We appreciate the reviewer for this insightful suggestion, which helped us further probe the bottleneck of PwS.
> > >
> > > We appreciate the reviewer for this insightful suggestion, which helped us further probe the bottleneck of PwS.
> > >
> > > Following your idea, we modified the **system prompt** (for PwS) to more explicitly regularize the output format: we now instruct the model, in the *Thought* part of its response, to (i) recognize and list the available objects and (ii) explicitly generate executable actions before outputting the final action. We then re-ran experiments on ALFWorld and SciWorld with 3 non-thinking models and 1 thinking model (DeepSeek-R1). The results are:
> > >
> > > **ALFWorld**
> > >
> > > | Model           | Prompt in Paper | Alternative Prompt |
> > > |-----------------|:---------------:|:------------------:|
> > > | DeepSeek-V3     | 7.0             | 9.0                |
> > > | Llama-4-Scout   | 12.0            | 9.0                |
> > > | Kimi-K2         | 45.0            | 47.0               |
> > > | DeepSeek-R1     | 14.0            | 8.0                |
> > >
> > > **SciWorld**
> > >
> > > | Model           | Prompt in Paper | Alternative Prompt |
> > > |-----------------|:---------------:|:------------------:|
> > > | DeepSeek-V3     | 25.5            | 14.9               |
> > > | Llama-4-Scout   | 41.4            | 29.7               |
> > > | Kimi-K2         | 52.9            | 52.8               |
> > > | DeepSeek-R1     | 46.9            | 51.5               |
> > >
> > > From these results, we observe that making schema instantiation more explicit in the prompt **does not yield consistent improvements**:
> > >
> > > - In **ALFWorld**, performance changes are small and mixed.
> > > - In **SciWorld**, the modified prompt often **degrades performance for non-thinking models** (DeepSeek-V3, Llama-4-Scout), has almost no effect on Kimi-K2, and **slightly improves DeepSeek-R1**.
> > >
> > > Qualitatively, with the alternative prompt, the thinking model (DeepSeek-R1) produces more comprehensive plans and a more detailed analysis of task progress. However, when it comes to recognizing available objects and instantiating templates, its reasoning traces are **highly similar to those of non-thinking models**. In particular, the model appears:
> > >
> > > - **Unaware of which parts of the pipeline require deeper reasoning** (e.g., we rarely observe reasoning steps that verify its instantiated actions), and
> > > - **Overconfident in its judgments about schema instantiation**, leading to similar failure patterns despite more verbose “thinking”.
> > >
> > > This suggests that merely encouraging explicit reasoning about schema instantiation at the **prompt level** does not fully address the bottleneck. Instead, we hypothesize that:
> > >
> > > Beyond heavy post-training on tool usage, **targeted training on environment sensing and schema grounding** (e.g., accurately detecting objects, affordances, and valid action instantiations in unfamiliar environments) may be crucial for building robust long-horizon agents under the PwS paradigm.
> > >
> > > ***We sincerely appreciate your valuable advice and hope that our response will assist you in raising your score. Thank you once again!***

---

> > > > ### Author Response · Authors · 2025-11-25
> > > >
> > > > Dear Reviewer 4W3y,
> > > >
> > > > As the rebuttal period is drawing to a close, we would be very grateful for any further comments or suggestions you might have on our submission. We hope that our additional experiments and clarifications have addressed your concerns and contributed positively to your assessment of the paper.
> > > >
> > > > Sincerely,
> > > >
> > > > The authors of submission 7312

---

### Author Response · Authors · 2025-12-01
**Summary of Previous Rebuttal Discussions**

Dear Area Chair,

Thank you very much for your time and effort in handling our submission to ICLR 2026. Given the increased workload introduced by the new reviewing policy, we provide below a concise summary of our discussions with the reviewers and the corresponding changes made to the paper.


---

### Reviewer 4W3y
**Original score:** 4

**Final score:** 4

**Score change:** Reviewer indicated **willingness to increase the score** in the initial review.

Reviewer 4W3y appreciated our alternative action representation, as well as the experiments and figures that helped illustrate our ideas. The main remaining concerns were:

1. The need for additional experiments to further verify the effectiveness of the proposed action representation.
2. A more explicit quantitative model of our Cognitive Bandwidth Hypothesis.

**Our response and changes:**

- We presented our current  trail of quantitative modeling of Cognitive Bandwidth in detail to the reviewer.
- We selected four representative models and conducted additional experiments to evaluate the proposed action representation under this framework.
- These new results are, in our view, both address the reviewer’s concerns and provide additional insight into our claims.

---

### Reviewer cDAS
**Original score:** 6

**Final score:** 6

**Score change:** No further response from the reviewer.

Reviewer cDAS’s concerns focused on three aspects:

1. Whether BabyAI would be a more suitable playground for comparing action representations.
2. The reasons behind performance differences between WebShop and ALFWorld.
3. A desire for more direct and explicit conclusions.

**Our response and changes:**

- For (1), we ran action representation comparisons on BabyAI using four representative models. The results further support and validate our main arguments.
- For (2), we added a more detailed analysis explaining the performance discrepancies between WebShop and ALFWorld.
- For (3), we refine the final analysis to present our key takeaways more directly and explicitly.

We believe these changes substantially address Reviewer cDAS’s concerns.

---

### Reviewer J1fi
**Original score:** 2

**Final score:** 6

**Score change: Increased from 2 to 6 on Nov 25, 09:43.**

Reviewer J1fi’s concerns were mainly about figure captions and experimental details. In particular, the reviewer requested:

- Clearer, token-level comparisons for context length differences between action representations.
- More detailed descriptions of model configurations and experimental setups (model usage, prompts).

**Our response and changes:**

- We provided token-level comparisons as requested.
- We added detailed descriptions of model usage, prompt design, and experimental settings.

**After these clarifications, the reviewer raised the score from 2 to 6, before the bug was revealed.**

---

Overall, we are grateful for the reviewers’ careful reading and constructive feedback. We feel that the reviewers recognize the contribution of our work, and that **their comments have helped us improve the clarity, empirical support, and impact of the paper, rather than fundamentally challenging its core ideas.**

**We hope this summary is helpful for your decision-making.**

Sincerely,
**Authors of Submission 7312**

---

### Meta-Review · Area_Chair_S4o7 · 2026-01-06

**Summary:**

This paper proposes Planning with Schemas (PwS), aiming to address the limitations of Planning with Actions (PwA) when facing long-horizon action-space scaling challenges in LLM agent tasks. There are several concerns and questions raised. The major common concern is that the performance compared to PwA in low-to-medium complexity settings is suboptimal. Others include lacking quantifiable metrics or probes for the postulated bandwidth or load variables. Additional issues include potential noise and dependence in behavioral comparison analysis (Reviewer 4W3y), robustness concerns about the abstraction practice of using [obj], and missing more explicit prompting design that specifically augments the SI stage. Reviewer cDAS questions the missing PwA results on TextCraft and further suggests another benchmark, BabyAI, and is also confused about the relationship between action paradigm choice and environment.
Reviewer J1fi has concerns about the weak rationale for the cognitive-load claims and other presentation clarity issues.

In the rebuttal, the authors provided a detailed response that largely addresses many concerns. However, the major common issue regarding suboptimal performance in low-to-medium complexity environments, including the newly added BabyAI results, remains outstanding. This limits the applicability of PwS when deployed in real, diverse difficulty-level environments.

**Reviewer Concerns:**

For Reviewer 4W3y, I believe the concerns are somewhat alleviated regarding the lack of quantifiable metrics or probes for the postulated bandwidth or load variables, robustness of the abstraction practice of using [obj], and the missing more explicit prompting strategies. However, the concerns about performance in low-to-medium complexity environments still remain.

Similarly for Reviewer cDAS, the inferior performance of PwS compared to PwA in ALFWorld is still outstanding. While other questions, I believe, have been clarified.

For Reviewer J1fi, I believe the rebuttal largely addresses the presentation issues. However, despite the rating score being raised to 6 by the reviewer, I have a follow-up question regarding the experimental setup for ALFWorld. It appears that only a 100-sample subset was used for ALFWorld. I am still unclear why the full ALFWorld benchmark cannot be used for testing.

**Reviewer Scores:**

Reviewer J1fi has raised the score to 6 after discussion. However, I think Reviewer 4W3y and Reviewer cDAS would likely maintain their scores, as the concern about the suboptimal performance of PwS has not been fully addressed.

---

### Decision · Program_Chairs · 2026-01-26

Reject